# A new methodology for upscaling semi-submersible platforms for floating offshore wind turbines

**Kaylie L. Roach, Matthew A. Lackner, and James F. Manwell**

Mechanical Engineering, University of Massachusetts Amherst, Amherst, MA, 01003, USA

**Correspondence:** Matthew A. Lackner (lackner@ecs.umass.edu)

**Abstract.** This paper presents a new upscaling methodology for semi-submersible floating offshore wind turbine platforms. The size and power rating of offshore wind turbines have been growing in recent years, with modern wind turbines rated at 10–18 MW in contrast with 2–5 MW in 2010. It is not apparent how much further wind turbine size can be increased before it is unjustified. Scaling relations are a useful method for analyzing wind turbine designs to understand the mass, load, and cost increases with size. Scaling relations currently do not exist but are needed for floating offshore platforms to understand how the technical and economic development of floating offshore wind energy may develop with increasing turbine size. In this paper, a hydrodynamic model has been developed to capture the key platform response in pitch. The hydrodynamic model is validated using OpenFAST, a high-fidelity offshore wind turbine simulation software. An upscaling methodology is then applied to two semi-submersible case studies of reference systems (the Offshore Code Comparison Collaboration Continuation (OC4) 5 MW and the International Energy Agency (IEA) 15 MW). For each case study, the platform pitch angle at rated wind turbine thrust is constrained to a specified value. The results show that platform dimensions scale to a factor of 0.75, and the platform steel mass scales to a factor of 1.5 when the wall thickness is kept constant. This study is the first to develop generalized upscaling relations that can be used for other triangular semi-submersible platforms that have three outer columns with the turbine mounted at the center of the system. This is in contrast with other studies that upscale a specific design to a larger power rating. This upscaling methodology provides new insight into trends for semi-submersible platform upscaling as turbine size increases.

## 1 Introduction

Offshore wind energy development continues to accelerate, and until now most offshore wind installations have used fixed-bottom support structures (Musial et al., 2022). Offshore wind turbines are now planned for areas with deeper water depths including the coastlines of California (Speer et al., 2016; Beiter et al., 2020), Japan (Yoshimoto et al., 2013), and Europe (Ågotnes et al., 2013), where floating offshore wind turbines (FOWTs) are needed (Musial et al., 2016; Jonkman and Matha, 2011). Floating platforms have been designed and deployed in pilot projects such as the Fukushima FORWARD project in Japan (Karimirad, 2014; Fukushima Offshore Wind Consortium, 2022; Kikuchi and Ishihara, 2019a), the Hywind project in Scotland (Karimirad, 2014; Skaare et al., 2006; Equinor, 2022), and the WindFloat deployments in Portugal (Karimirad, 2014; Principle Power, 2022; Beaubouef, 2022), and Scotland (Memija, 2023) with turbine power ratings of 2–9.5 MW, while others are planned for the US in California (Lackner, 2021; California Energy Commission, 2023) and Maine (State of Maine Governor's Energy Office, 2020).

Offshore wind turbine size and capacity have been growing rapidly over the past 10 years as well. Modern offshore wind turbines designed by General Electric (GE), Siemens Gamesa, and Vestas have ratings of 10–18 MW with blade diameters exceeding 200 m (Siemens Gamesa, 2023; Vestas, 2023; GE Renewable Energy, 2023). GE has only recently announced their 18 MW turbine design (Buljan, 2023). In contrast, in 2010 offshore turbines had power ratings be-

tween 2–5 MW, and blade diameters were 75–125 m (Musial et al., 2022). Even larger designs are likely to be developed in the future, with researchers even investigating a 50 MW turbine (Yao et al., 2021).

While the industry is clearly trending towards larger wind turbines, the classical "square–cube" law dictates that the per MW capital cost of a wind turbine increases with turbine size due to the mass increasing more quickly than the rated power (Manwell et al., 2009). However, looking at data of historic wind turbines, the cost does not scale with the mass because of technological innovations over time (Jamieson, 2018). Also, the industry trend towards larger offshore wind turbines minimizes the number of installed units in a wind farm for a given total capacity, which is motivated by the large per unit cost (including the foundation, installation, electrical interconnection, and maintenance visits at sea). Offshore wind levelized cost of energy (LCOE) is still about twice as much as onshore wind on average, but as turbine size has increased, LCOE has decreased significantly over time (Thresher et al., 2008; Beiter et al., 2016). As offshore wind energy development continues, it is important to understand if even larger turbines can continue to reduce the LCOE of offshore wind farms or if there is an upper limit to the cost effectiveness and practicality of upscaling.

The process of evaluating a wind turbine design with increasing scale is referred to as upscaling. Classical upscaling methods can be used to project the power, size, mass, forces, moments, costs, and other properties of an upscaled turbine based on a turbine of a smaller size (Manwell et al., 2009). Upscaling methods are discussed further in Sect. 2. As wind turbines are rapidly increasing in power rating, research is needed to understand how the design characteristics of FOWT platforms, including the physical dimensions, mass, cost, and dynamic behavior, change with respect to the increased turbine size.

This paper aims to model and analyze semi-submersible FOWT platform design characteristics and system dynamics to provide insight into the technological development of FOWT systems with larger power ratings. The objective is to develop general scaling trends, which characterize the mass, dimensions, and dynamics of the semi-submersible FOWT platform subject to constraints on the system stability as a turbine is upscaled. To achieve this objective, a new upscaling methodology for floating platforms is developed based on a hydrodynamic model that captures the key platform responses in pitch. The hydrodynamic model is validated using OpenFAST, a high-fidelity offshore wind turbine simulation software (Jonkman, 2019; National Renewable Energy Laboratory, 2020). The methodology is then applied using two semi-submersible case studies, in which the platform pitch angle at rated wind turbine thrust is constrained to a constant value. Other researchers have upscaled specific semi-submersible platforms (Kikuchi and Ishihara, 2020; George, 2014; Leimeister et al., 2016; Ju et al., 2020). This study is the first to develop generalized upscaling rela-

tions for triangular semi-submersible FOWT platforms with three outer columns and a centrally mounted turbine. Ideally, this research would also be conducted with other types of semi-submersible designs, as well as other FOWT designs (spar, tension leg platform). These upscaling relations can provide new insight into design trends for triangular semi-submersible platforms as turbine size increases. Additionally, the paper identifies key underlying physics behind the semi-submersible upscaling relations.

This paper is organized as follows: Sect. 2 provides a literature review. Section 3 describes the methods used in this research study, including the hydrodynamic modeling of floating offshore platforms, the semi-submersible case studies, the model validation, and upscaling methodology. Section 4 presents the upscaling results, as well as a new analytical model for FOWT upscaling and parameter sensitivity studies. Finally, Sect. 5 summarizes the research findings and future work.

## 2    Background

Classical upscaling relations have been developed for a wind turbine with geometric and aerodynamic similarity (Manwell et al., 2009; Sieros et al., 2012; Ashuri, 2012). The general form of the scaling relations is shown in Eq. (1). The upscaled parameter (denoted with subscript 2) depends on the ratio of the upscaled to the original rotor radius ($R$), original parameter size (denoted with subscript 1), and the scale dependence power $\alpha$. Table 1 shows the scaling relations for power, forces, weight, moments, stresses, and resonances for a wind turbine (Manwell et al., 2009).

$$\frac{\text{Parameter}_1}{\text{Parameter}_2} = \left(\frac{\text{Radius}_1}{\text{Radius}_2}\right)^{\alpha} = R^{\alpha} \tag{1}$$

The rotor power is related to the scaling factor squared ($R^2$) because it is proportional to the rotor swept area. The weight of the wind turbine rotor increases with $R^3$ because of the volumetric upscaling with geometric similarity (Manwell et al., 2009). This square–cube law therefore implies that mass will increase more quickly than rated power as a turbine is upscaled, which would seem to argue against increasing the turbine size. The aerodynamic stresses are independent of rotor size, while stresses due to the blade weight increase in proportion to the rotor radius and may eventually drive the design loads for an upscaled rotor.

Historical data from wind turbines of different sizes can also be used to understand upscaling trends. For example, historical data indicate that the rotor mass has increased to the power of between 2 and 2.5, not the cubic power of the square–cube law (Fig. 1) (Jamieson, 2018). This smaller value for the scaling exponent is primarily due to technological innovation, such as new materials and improved manufacturing, in newer designs that are usually larger in size (Jamieson, 2018; Shields et al., 2021).

**Table 1.** Classical scaling relations (Manwell et al., 2009).

| Quantity | Symbol | Relation | Scale dependence |
|---|---|---|---|
| Power | $P$ | $P_1/P_2 = (R_1/R_2)^2$ | $\sim R^2$ |
| Torque | $Q$ | $Q_1/Q_2 = (R_1/R_2)^3$ | $\sim R^3$ |
| Thrust | Th | $Th_1/Th_2 = (R_1/R_2)^2$ | $\sim R^2$ |
| Rotational speed | $\Omega$ | $\Omega_1/\Omega_2 = (R_1/R_2)^1$ | $\sim R^{-1}$ |
| Weight | $W$ | $W_1/W_2 = (R_1/R_2)^3$ | $\sim R^3$ |
| Aerodynamic moments | $M_A$ | $M_{A,1}/M_{A,2} = (R_1/R_2)^3$ | $\sim R^3$ |
| Centrifugal forces | $F_c$ | $F_{c,1}/F_{c,2} = (R_1/R_2)^2$ | $\sim R^2$ |
| Gravitational stress | $\sigma_g$ | $\sigma_{g,1}/\sigma_{g,2} = (R_1/R_2)^1$ | $\sim R^1$ |
| Aerodynamic stress | $\sigma_A$ | $\sigma_{A,1}/\sigma_{A,2} = (R_1/R_2)^0 = 1$ | $\sim R^0$ |
| Centrifugal stress | $\sigma_c$ | $\sigma_{c,1}/\sigma_{c,2} = (R_1/R_2)^0 = 1$ | $\sim R^0$ |
| Natural frequency | $\omega$ | $\omega_{n,1}/\omega_{n,2} = (R_1/R_2)^1$ | $\sim R^{-1}$ |
| Excitation | $\Omega/\omega$ | $(\Omega_1/\omega_{n,1})/(\Omega_2/\omega_{n,2}) = (R_1/R_2)^0 = 1$ | $\sim R^0$ |

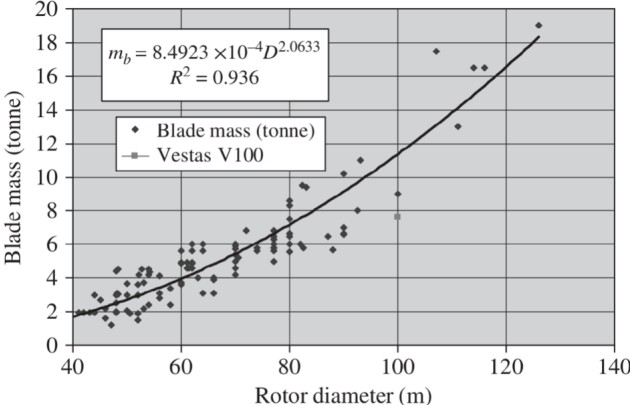

$m_b = 8.4923 \times 10^{-4} D^{2.0633}$
$R^2 = 0.936$

Blade mass (tonne)
Vestas V100

**Figure 1.** Blade mass scaling based on data (Jamieson, 2018).

As wind turbines become larger, the mass and aerodynamic forces increase as well, and so the floating platform that supports the turbine must grow to ensure a stable system. Several researchers have studied floating platform upscaling for a specific case study (Kikuchi and Ishihara, 2020; Leimeister et al., 2016; George, 2014; Ju et al., 2020), including multiple studies of semi-submersible FOWT platforms (Kikuchi and Ishihara, 2019a; George, 2014; Leimeister et al., 2016). George (2014) upscales the 5 MW Offshore Code Comparison Collaboration Continuation (OC4) semi-submersible FOWT to 7.5 and 10 MW (Robertson et al., 2014). Leimeister (2016) also upscale the 5 MW OC4 reference FOWT platform to 7.5 and 10 MW. Kikuchi and Ishihara (2019a) upscale the 2 MW Fukushima FORWARD semi-submersible FOWT to 5 and 10 MW (Fukushima Offshore Wind Consortium, 2022). Leimeister et al. (2016) upscale all platform parameters and then check the static pitch of the turbine at rated wind speed to iteratively adjust parameters as needed. Both George (2014) and Kikuchi and Ishihara (2019b) keep the draft constant due to constraints of harbor depth. The other parameters are scaled, and the static pitch is evaluated; the design is iterated until the static pitch matches the original design. Each of these three studies finds that it is technically and economically feasible to upscale the semi-submersible system. Leimeister et al. (2016) find that the upscaled system also had to be designed for the heave natural period and recommend having different scaling factors for different parts of the platform.

Wu and Kim (2021) have developed a methodology for upscaling a FOWT turbine and semi-submersible platform by using the 5 MW OC4 and 15 MW International Energy Agency (IEA) semi-submersible systems. The central column diameter is set to be equal to the tower diameter, and a guess is made for the scale factor, which is applied to the column radius and distance between columns. The buoyancy is calculated, and the ballast mass is set to match the total weight with the buoyancy. The scaling factor for the distance between columns and column radius is adjusted iteratively until the desired platform pitch angle is reached. Additionally, the same methodology is followed while keeping the column radius constant and only increasing the distance between the columns.

When upscaling a FOWT, specific load cases are typically used to constrain the design and ensure acceptable stability and dynamics. Load cases at rated wind speed often govern the extreme loads of FOWT systems (Kikuchi and Ishihara, 2019a; George, 2014; Leimeister et al., 2016; Wu and Kim, 2021; de Souza and Bachynski-Polić, 2022). De Souza and Bachynski-Polić (2022) study the behavior of a large spar FOWT and find that the extreme loads are governed by the rated wind speed cases rather than the extreme wind and sea state cases.

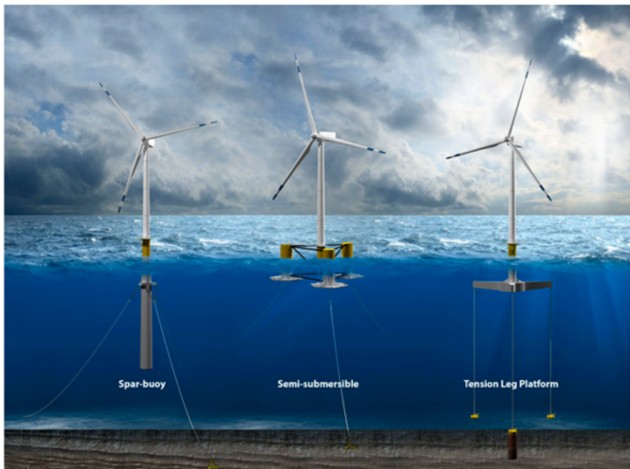

**Figure 2.** Types of floating offshore wind platforms (illustration by Joshua Bauer, National Renewable Energy Laboratory – NREL) (Speer et al., 2016).

## 3   Methodology

In this section, a hydrodynamic model for FOWT platforms is presented, which can be used to assess the static stability and natural period of a platform based on the geometry. The hydrodynamic model is validated using OpenFAST. This model is then used in two upscaling case studies, which are carried out for two semi-submersible platforms.

### 3.1   Hydrodynamic modeling of floating platforms

FOWT platforms stabilize the offshore wind turbine system, allowing the turbine to produce power while floating in the water. Figure 2 shows the three primary FOWT platform types: spar, semi-submersible, and tension leg platform (Speer et al., 2016). Floating platforms can be stabilized by ballast, buoyancy, moorings, or a combination (Speer et al., 2016; Wang et al., 2010; Thiagarajan and Dagher, 2014; Karimirad, 2014). This study focuses on semi-submersible FOWT platforms, which are primarily stabilized by the large waterplane area of the offset columns, with a wider spread adding more stability.

OpenFAST is a simulation tool developed by the National Renewable Energy Laboratory (NREL) used to evaluate offshore wind turbines (National Renewable Energy Laboratory, 2020; Jonkman, 2019). The aerodynamics, hydrodynamics, elastodynamics, and system controls are all incorporated into a coupled simulation. OpenFAST is widely used in academia and industry for wind turbine modeling and simulation.

The stability of a FOWT can be characterized using the hydrodynamic loading and response. Eq. (2), known as the Cummins equation, is the equation of motion for an offshore platform in water with 6 degrees of freedom (Duarte et al., 2014; TU Delft, 2006; Jonkman, 2007). $M_{\text{ii}}$ is the

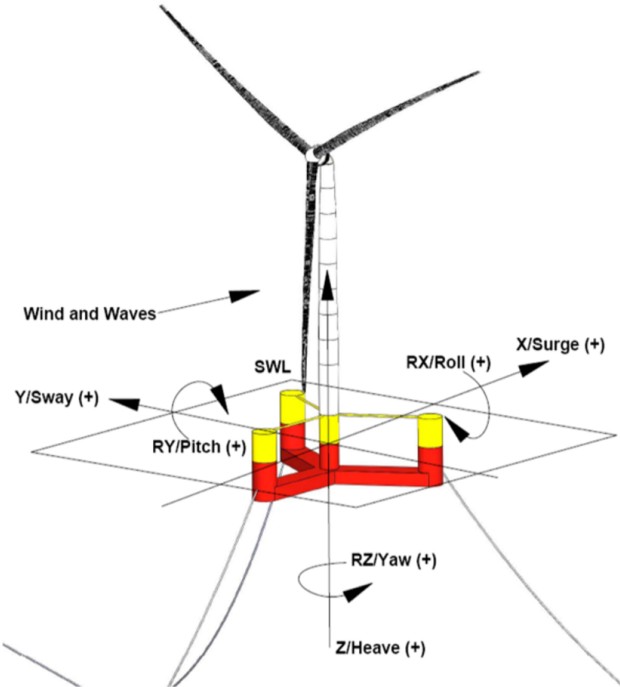

**Figure 3.** FOWT platform degrees of freedom on the International Energy Agency 15 MW system (Allen et al., 2020).

mass or mass moment of the inertia term, $A_{\text{ii}}$ is the added mass coefficient term, $K_{\text{ii}}$ is the retardation matrix, and $C_{\text{ii}}$ is the stiffness matrix. The platform acceleration, velocity, and displacements are represented by $\ddot{q}^{\text{tot}}$, $\dot{q}^{\text{tot}}$, and $q^{\text{tot}}$, respectively; $F_i^{\text{waves}}$ is the external wave loading; $F_i^{\text{rotor}}$ is the force of the wind turbine acting on the floating platform; and $h_i$ is the moment arm of $F_i^{\text{rotor}}$ for rotational platform degrees of freedom. The 6 degrees of freedom are labeled with $i = 1, 2, \ldots, 6$ and correspond to surge, sway, heave, roll, pitch, and yaw. Figure 3 shows the FOWT coordinate system (Sebastian and Lackner, 2012).

$$(M_{\text{ii}} + A_{\text{ii}})\ddot{q}^{\text{tot}} + \int_0^T K_{\text{ii}}(t - \tau)\,\dot{q}^{\text{tot}}\tau\,\mathrm{d}\tau + C_{\text{ii}}q^{\text{tot}}$$
$$= F_i^{\text{waves}} + F_i^{\text{rotor}} \cdot h_i \tag{2}$$

When the platform is in static equilibrium, the acceleration and velocity terms are 0; ignoring the wave forcing leaves only the hydrostatic stiffness term balancing the aerodynamic forces and moments $C_{\text{ii}}q^{\text{tot}} = F_i^{\text{rotor}} \cdot h_i$. The stiffness term, $C_{\text{ii}}$, is comprised of both platform stiffness, $C_{\text{ii}}^{\text{hydrostatics}}$, and mooring line stiffness, $C_{\text{ii}}^{\text{lines}}$. This study focuses on the stiffness contributions from the platform rather than the mooring lines. The mooring lines provide a restoring force in the surge (1,1), sway (2,2), and yaw (6,6) degrees of freedom, but this study focuses primarily on the pitch (5,5) degree of freedom (Delhommeau, 1993). For the pitch degree of freedom, $h_i$ is the distance from the system center of mass to the rotor hub.

In OpenFAST, the hydrostatic stiffness matrix, $C_{ii}^{\text{hydrostatics}}$, is defined using only the waterplane area and center of buoyancy; the center of mass is calculated separately (Jonkman, 2007). However, the hydrostatic stiffness of a platform has contributions from both gravity and buoyancy in this study, which is traditional in the field of naval architecture, and is used in the pitch angle (Eq. 4 below). Equation (3) shows the hydrostatic stiffness matrix, $C_{ii}^{\text{hydrostatics}}$, for an offshore platform (Delhommeau, 1993). The displaced volume is $V_{\text{disp}}$, the center of buoyancy is $B$, and the center of mass is CM. The matrix is symmetric and has non-zero components including (3,3), (4,4), (5,5), (3,4), (3,5), and (4,5), corresponding to the heave (3,3), roll (4,4), and pitch (5,5) degrees of freedom:

$$C_{ij}^{\text{hydrostatics}} = \begin{bmatrix} 0 & 0 & 0 & 0 & 0 & 0 \\ 0 & 0 & 0 & 0 & 0 & 0 \\ 0 & 0 & C_{33} & C_{34} & C_{35} & 0 \\ 0 & 0 & C_{43} & C_{44} & C_{45} & 0 \\ 0 & 0 & C_{53} & C_{54} & C_{55} & 0 \\ 0 & 0 & 0 & 0 & 0 & 0 \end{bmatrix}, \quad (3)$$

with

$$C_{33} = \rho g W_0,$$

$$C_{44} = \rho g \int\int_{W_0} Y^2 \mathrm{d}W + \rho g V_{\text{disp}}(B - \text{CM}, )$$

$$C_{55} = \rho g \int\int_{W_0} X^2 \mathrm{d}W + \rho g V_{\text{disp}}(B - \text{CM}, )$$

$$C_{34} = C_{43} = \rho g \int\int_{W_0} Y \mathrm{d}W,$$

$$C_{35} = C_{53} = -\rho g \int\int_{W_0} X \mathrm{d}W,$$

$$C_{45} = C_{54} = -\rho g \int\int_{W_0} XY \mathrm{d}W.$$

The platform has a non-zero mean pitch angle during normal operation due to aerodynamic forces. The static platform pitch angle at rated thrust (maximum thrust condition) can be calculated using Eq. (4) based on the thrust at rated wind speed, $F_5^{\text{rotor}}$; height from rotor nacelle assembly to the waterline, $h_{\text{hub}}$; and pitch stiffness, $C_{55}^{\text{hydrostatics}}$.

$$\theta_p = \frac{F_5^{\text{rotor}} h_{\text{hub}}}{C_{55}^{\text{hydrostatics}}} \quad (4)$$

The natural period for offshore structures with catenary moorings is typically over 100 s in surge, sway, and yaw and over 20 s in heave, roll, and pitch (Det Norske Veritas Germanischer Lloyd, 2017). The natural period of the system is designed to be outside the dominant period range of the wave climate so that the structure is not excited by the ocean waves. The natural period for a moored structure is approximately given by Eq. (5) (Det Norske Veritas Germanischer

**Table 2.** The Offshore Code Comparison Collaboration Continuation platform properties (Robertson et al., 2014).

| | | |
|---|---|---|
| Draft | 20 | m |
| Heave plate height ($H_{\text{hp}}$) | 6 | m |
| Freeboard | 12 | m |
| Spacing between columns ($\text{Dist}_{\text{cc}}$) | 50 | m |
| Column radius ($\text{Rad}_{\text{col}}$) | 6 | m |
| Heave plate radius ($\text{Rad}_{\text{hp}}$) | 12 | m |
| Ballast density | 1025 | kg m$^{-3}$ |
| Platform mass including ballast | 1.33E+07 | kg |
| $I_{\text{platform}}$ about $\text{CM}_{\text{platform}}$ in pitch | 6.827E+09 | kg m$^2$ |
| Platform $\text{CM}_{\text{system}}$ below waterline | −13.46 | m |

**Table 3.** The 5 MW reference turbine properties (Jonkman et al., 2009).

| | | |
|---|---|---|
| Rating | 5 | MW |
| Rotor radius | 63 | m |
| Hub height | 90 | m |
| Rated wind speed | 11.4 | m s$^{-1}$ |
| Rotor mass | 110 000 | kg |
| Nacelle mass | 240 000 | kg |
| Tower mass | 249 718 | kg |
| Specific power | 401 | W m$^{-2}$ |

Lloyd, 2017; Kikuchi and Ishihara, 2020). This research includes hydrostatic stiffness but not mooring line stiffness; a mooring line sensitivity study is presented in Sect. 4.6.1.

$$T_i = 2\pi \sqrt{\frac{M_{\text{ii}} + A_{\text{ii}}}{C_{\text{ii}}^{\text{hydrostatics}} + C_{\text{ii}}^{\text{lines}}}} \quad (5)$$

## 3.2 Case study for semi-submersible models

Two semi-submersible platforms are used as case studies for upscaling. Reference FOWT systems developed by both NREL and the International Energy Agency (IEA) are selected: the OC4 5 MW semi-submersible (Robertson et al., 2014) and the IEA 15 MW semi-submersible (Allen et al., 2020). In Sect. 3.4, an upscaling methodology for the floating platforms is presented, which is then applied to these two case studies.

### 3.2.1 OC4 semi-submersible model

The 5 MW OC4 is a semi-submersible platform with three outer columns and one central column below the tower, connected with cross-braces (Fig. 4). The properties are shown in Table 2. Seawater ballast is used within the three outer columns, with the heave plates filled and the upper part of the column partially filled. The 5 MW reference turbine has a 63 m radius with a rated wind speed of 11.4 m s$^{-1}$. The turbine properties are summarized in Table 3.

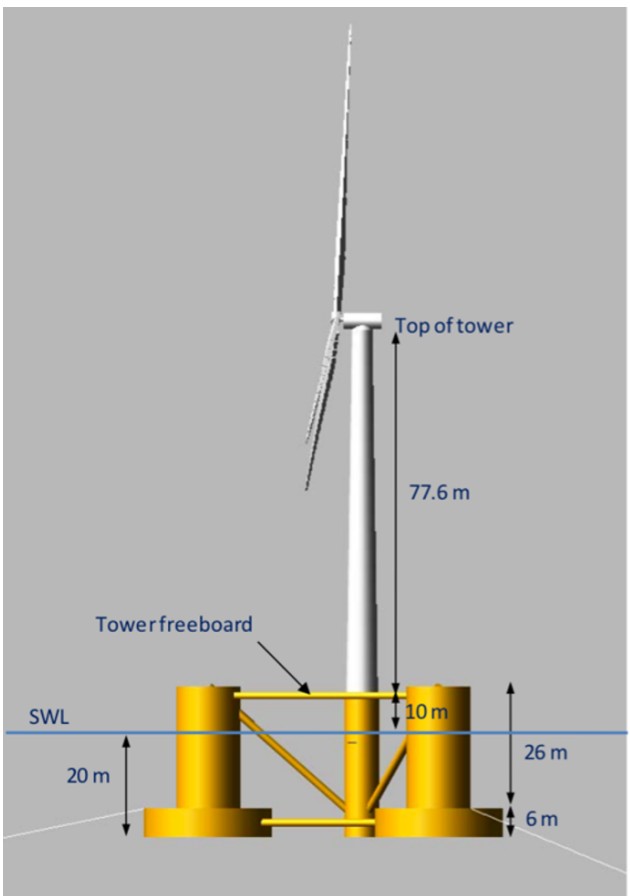

**Figure 4.** OC4 platform dimensions (Robertson et al., 2014).

The wind turbine is upscaled to 10, 15, and 20 MW using aerodynamic similarity and by holding the specific power (Sp) constant. Specific power is defined as the rated power, $P_R$, divided by the rotor swept area and is reported in units of $W\,m^{-2}$ in Eq. (6).

$$Sp = \frac{P_R}{\pi R^2} \tag{6}$$

The specific power of the original OC4 turbine is $401\,W\,m^{-2}$. The rotor diameter is calculated for each upscaled turbine rating (Table 4). The tower mass is upscaled by a factor of 2, which is based on the upscaling trends of the reference turbine towers. The rotor nacelle assembly (RNA) mass is upscaled by a factor of 2.2 based on the literature on upscaling trends that account for technological advancement (Jamieson, 2018). The hub height is calculated assuming that there is a 30 m gap between the bottom of the rotor plane and the waterline. The methodology for upscaling the platform for each turbine model is presented in Sect. 3.4.

**Table 4.** Rotor radius of upscaled OC4 turbines.

| Power rating (MW) | 10 | 15 | 20 |
|---|---|---|---|
| Rotor radius (m) | 89 | 109 | 126 |
| Hub height (m) | 119 | 139 | 156 |

**Table 5.** The IEA 15 MW semi-submersible platform properties (Allen et al., 2020).

| | | |
|---|---|---|
| Draft | 20 | m |
| Freeboard | 15 | m |
| Distance between columns ($Dist_{cc}$) | 89.63 | m |
| Radius of upper columns ($Rad_{col}$) | 6.25 | m |
| Pontoon height ($H_{pon}$) | 7 | m |
| Platform mass including ballast | 1.78E+07 | kg |
| Seawater ballast mass | 1.13E+07 | kg |
| Iron ore ballast mass | 4.80E+06 | kg |
| $I_{platform}$ about $CM_{platform}$ in pitch | 1.251E+10 | $kg\,m^2$ |
| $CM_{platform}$ below waterline | −14.94 | m |

### 3.2.2    IEA semi-submersible model

The IEA 15 MW turbine was designed with both a semi-submersible platform and a monopile (Allen et al., 2020; Gaertner et al., 2020). The 15 MW IEA semi-submersible platform has three outer columns and one central column to support the turbine (Fig. 3). The 20 m draft is the same as the OC4 semi-submersible. Semi-submersible platforms have a relatively shallow draft compared to a spar platform, and this is prioritized for the 15 MW design which has a 20 m draft. One noticeable difference between the OC4 and IEA designs is the pontoons between the three outer columns, instead of heave plates. The IEA 15 MW semi-submersible platform properties are shown in Table 5.

The 15 MW reference turbine has a rotor radius of 120 m and a rated wind speed of $10.59\,m\,s^{-1}$. The turbine has a lower specific power of $332\,W\,m^{-2}$ compared to the 5 MW reference turbine ($401\,W\,m^{-2}$) because of the lower rated wind speed. The turbine properties are summarized in Table 6. The 15 MW IEA semi-submersible wind turbine is upscaled to 20, 25, and 30 MW. The rotor diameters of the three upscaled turbines are shown in Table 7. Again, the tower mass is upscaled by a factor of 2, and the RNA mass is upscaled by a factor of 2.2. The hub height is calculated assuming that there is a 30 m gap between the bottom of the rotor plane and the waterline.

### 3.3    Verification of the hydrodynamic model for case study turbines

The hydrodynamic model presented in Sect. 3.1 is validated by simulating the two case study reference turbines in OpenFAST (National Renewable Energy Laboratory, 2020; Jonkman and Buhl, 2005). OpenFAST is used to calculate the static platform pitch under steady, rated wind speed and

**Table 6.** The 15 MW reference turbine properties (Gaertner et al., 2020).

| Rated power | 15 | MW |
|---|---|---|
| Rotor radius | 120 | m |
| Hub height | 150 | m |
| Rated wind speed | 10.59 | $\mathrm{m\,s^{-1}}$ |
| Rotor mass | 3.85E+05 | kg |
| Nacelle mass | 6.31E+05 | kg |
| Tower mass | 1.26E+06 | kg |
| Sp | 332 | $\mathrm{W\,m^{-2}}$ |

**Table 7.** Rotor radius of upscaled IEA turbines.

| Power rating (MW) | 20 | 25 | 30 |
|---|---|---|---|
| Rotor radius (m) | 138 | 155 | 170 |
| Hub height (m) | 168 | 185 | 200 |

to calculate the pitch natural period ($T_n$) of the system. The OC4 semi-submersible platform result for platform pitch angle is shown in Fig. 5, which is estimated as 3.26°. The platform pitch value calculated using the presented hydrody-
5 namic model in Eq. (4) is 3.55°. Both platform pitch angles are relative to the waterline. The 9 % error is acceptable for the purpose of setting the platform pitch angle for upscaling, especially since the proposed model is much less computationally expensive than OpenFAST.
The natural period of the OC4 semi-submersible is evaluated in OpenFAST by using a free decay test (Fig. 6), with an initial platform pitch angle of 8°. Based on this test, the $T_n$ of the system is 25.5 s. The published $T_n$ is 27.0 s (Robertson et al., 2014). The pitch $T_n$ of the system calculated using
the hydrodynamic model in Eq. (5) is 24.2 s, with a 10 % error relative to the published value and a 5 % error relative to the $T_n$ found using OpenFAST. The error is likely due to second-order effects in OpenFAST that are not captured in the hydrodynamic model.
OpenFAST is also used to simulate the 15 MW IEA wind turbine with the semi-submersible platform. The static platform pitch angle is estimated as 3.6° at steady, rated wind speed (Fig. 7). The platform pitch angle found using the hydrodynamic model in Eq. (4) is 4.9°. This 36 % error in static
pitch angle as compared with the OpenFAST model may be due to limitations in what is known about the IEA 15 MW system. For instance, the system center of mass and moment of inertia are published for the 5 MW OC4 system but not for the IEA 15 MW system. The platform pitch angle from
the hydrodynamic model can be used as a relative rather than absolute pitch angle in order to constrain the upscaled turbine platform pitch angle. The pitch $T_n$ from the OpenFAST free decay test (Fig. 8) is estimated as 27.7 s, the published value is 29.5 s, and the result from the hydrodynamic model
is 28.6 s. The model has a 3 % error relative to the Open-FAST results and a 3 % error relative to the published value.

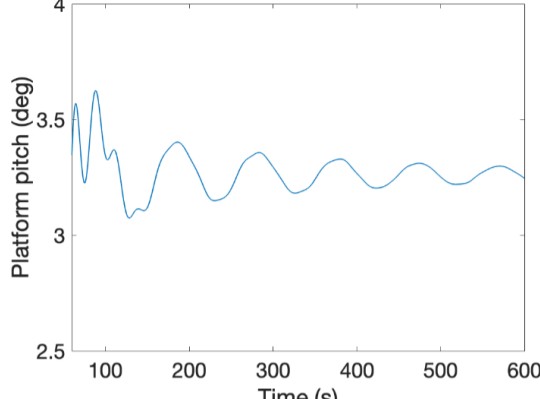

**Figure 5.** The OC4 platform pitch angle at rated wind speed using OpenFAST.

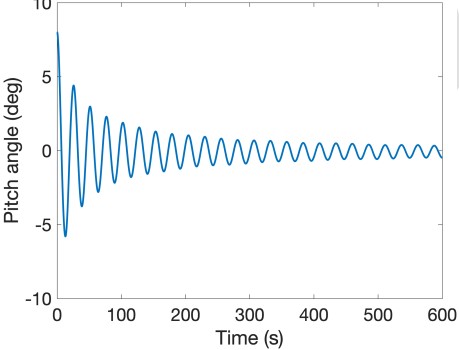

**Figure 6.** Free decay of the OC4 semi-submersible using Open-FAST.

The verification results are summarized in Table 8. The purpose of this verification was to confirm that the calculations were similar to both published values as well as OpenFAST simulations. The model could be further verified with other 40 simulation software or with data from FOWT pilot projects, but further validation is outside the scope of this paper.

## 3.4 Upscaling methodology

The semi-submersible platforms are upscaled by first upscaling the turbine to a higher power rating and then using the 45 following methodology for the platforms:

1. linearly increase the platform dimensions, specifically the column radii and spacing, with a scaling constant $\alpha$ (Eq. 1);

2. use the hydrodynamic model to find the static pitch 50 angle at rated wind speed (Eq. 4) and natural period (Eq. 5);

3. continue to increase the platform dimensions with the scaling constant $\alpha$ until the upscaled static pitch angle

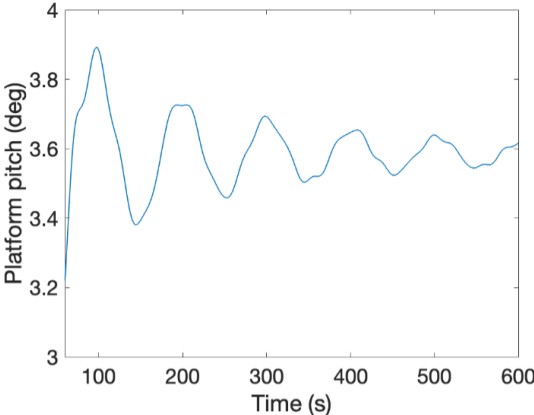

**Figure 7.** The IEA 15 MW platform pitch angle at rated wind speed using OpenFAST.

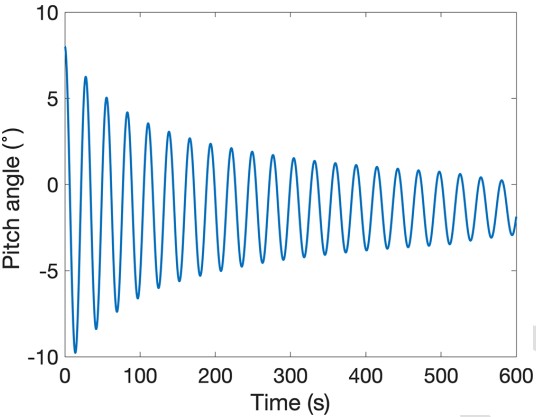

**Figure 8.** Free decay of the IEA 15 MW using OpenFAST.

is equivalent to the static pitch angle of the original case study.

This method is effectively a root-finding problem to determine the value of $\alpha$ that results in equal rated platform pitch angles for the baseline and upscaled turbines. While it may be possible to solve for a single $\alpha$ value analytically, the root-finding approach was selected because it allows us to see trends in the platform behavior. We can clearly see how the upscaling value of $\alpha$ results in a more conservative or less conservative design. The platform dimensions are upscaled uniformly with the scaling constant $\alpha$ in Eq. (1), which is increased from 0 to 2 in increments of 0.005. The wall thickness and clearance between the blade tip and the waterline are kept constant during upscaling. The 30 m clearance between the rotor and the waterline was chosen because the literature and industry trends show that the 30 m clearance is typical for offshore wind turbines to date (Robertson et al., 2014; Allen et al., 2020). The system mass, buoyancy, ballast mass, center of buoyancy, center of mass, static pitch stiffness, static pitch angle, and pitch natural period are cal-

**Table 8.** Model verification.

|  |  | OC4 5 MW | IEA 15 MW |
| --- | --- | --- | --- |
| Platform pitch angle (°) | Hydrodynamic model | 3.94 | 4.90 |
|  | OpenFAST | 3.26 | 3.60 |
| Pitch natural period (s) | Hydrodynamic model | 24.2 | 28.6 |
|  | OpenFAST | 25.5 | 27.7 |
|  | Published | 27.0 | 29.5 |

culated for each turbine size (10–30 MW) and $\alpha$ scaling constant. In this paper $CM_{system}$ includes the total system center of mass, including the turbine, tower, and platform. In contrast, $CM_{platform}$ is used for the platform center of mass, excluding the turbine and tower. The platform pitch stiffness is calculated using Eq. (7), which comes from $C_{55}$ in Eq. (3). The distance from one outer column to another is $Dist_{CC}$.

$$C_{55} = \rho g \left( V_{disp} \left( B - CM_{system} \right) + \frac{\pi}{4} (Rad_{cent})^4 \right.$$
$$\left. + \frac{3\pi}{4} (Rad_{col})^4 + 2\pi (Rad_{col})^2 \left( \frac{Dist_{cc}}{2} \right)^2 \right) \qquad (7)$$

The platform is upscaled until the platform pitch angle at rated wind speed matches the initial design platform pitch angle in Eq. (4). The pitch $T_n$ is calculated using Eq. (8) (derived from Eq. 5) to ensure that it is not in the predominant wave period range. The pitch $T_n$ of a semi-submersible platform should always be above 20 s (Det Norske Veritas Germanischer Lloyd, 2017). The added mass coefficient $C_A$ comes from the documentation for each semi-submersible case study (Robertson et al., 2014; Allen et al., 2020). The moment of inertia of the system is $I_{system}$, and the moment of inertia of the platform is $I_{platform}$.

$$T_{55} = \sqrt{\frac{I_{system} + C_A I_{platform}}{C_{55}}} \qquad (8)$$

### 3.4.1 OC4 semi-submersible upscaling method

The OC4 semi-submersible turbine is upscaled from 5 to 10, 15, and 20 MW. The OC4 platform draft is kept at a constant 20 m, and the wall thickness is kept constant at 6 cm. The ballast is seawater with a density of 1025 kg m$^{-2}$. The center of mass of the entire OC4 system is $-10$ m, while the center of mass ($CM_{platform}$) of the OC4 platform is $-13.46$ m.

The platform displaced volume is set using Eq. (9). The system buoyancy is equal to the mass of the displaced water. The platform steel mass is calculated with the upscaled dimensions, and the ballast mass is the difference between the buoyancy and the steel mass. The radius of the outer columns at the waterline is $Rad_{col}$, the height of the heave plate on the lower part of the columns is $H_{hp}$, the radius of the heave plate is $Rad_{hp}$, and the radius of the central column below the

tower is $\text{Rad}_{\text{center}}$.

$$V_{\text{disp}} = 3 \left( \pi (\text{Rad}_{\text{col}})^2 \left( \text{draft} - H_{\text{hp}} \right) + \pi \left( \text{Rad}_{\text{hp}} \right)^2 H_{\text{hp}} \right) + \pi (\text{Rad}_{\text{cent}})^2 \text{draft} \tag{9}$$

### 3.4.2 IEA semi-submersible upscaling method

The IEA 15 MW turbine is upscaled to 20, 25, and 30 MW. Since both the 15 MW IEA and the 5 MW OC4 had an equivalent draft of 20 m, the OC4 platform upscaling kept the draft constant. However, the IEA platform draft is increased with $\alpha$ for the larger turbines. The upscaled IEA platform wall thickness is kept at a constant 4.5 cm. The IEA platform has seawater ballast filling the pontoons and an iron ore ballast partially filling the columns. The iron ore ballast density is estimated to be 4300 kg m$^{-2}$. The center of buoyancy, center of mass of the platform, and center of mass of the entire system are calculated. The $\text{CM}_{\text{platform}}$ of the IEA 15 MW platform is $-15$ m, while the total $\text{CM}_{\text{system}}$ of the system is $-2.8$ m.

The displaced volume is calculated using Eq. (10). The platform steel mass, buoyancy mass, and ballast mass are calculated. The seawater ballast fills up the pontoon inner volume. The remaining ballast mass partially fills the three outer columns with iron ore. The column radius for this type of semi-submersible platform is $\text{Rad}_{\text{col}}$, and the radius of the central column is $\text{Rad}_{\text{center}}$; the pontoon length, width, and height are $L_{\text{pon}}$, $W_{\text{pon}}$, and $H_{\text{pon}}$, respectively.

$$V_{\text{disp}} = 3 \left( \pi (\text{Rad}_{\text{col}})^2 \text{draft} + L_{\text{pon}} \cot W_{\text{pon}} \cdot H_{\text{pon}} \right) + \pi (\text{Rad}_{\text{cent}})^2 \text{draft} \tag{10}$$

## 4  Results and discussion

In this section, the case study results using the methodology presented in Sect. 3 are analyzed. First, the results are given for the OC4 platform upscaling results (Sect. 4.1) and the IEA platform upscaling results (Sect. 4.2). The results from both case studies are compared with each other (Sect. 4.3) and then compared with similar studies from the literature (Sect. 4.4). An analytical model for semi-submersible platform upscaling is shown as a comparison to the iterative upscaling results (Sect. 4.5), and the sensitivity studies are presented (Sect. 4.6).

### 4.1  OC4 platform upscaling results

The platform pitch angle at rated thrust is plotted for each upscaling factor value in Fig. 9. As the $\alpha$ value increases, the platform dimensions increase, and the static platform pitch angle decreases. The platform pitch angle of the upscaled platforms matches the OC4 angle of 3.5° at an $\alpha$ of 0.75. Further investigation of the upscaling factor is shown in the analytical model for the semi-submersible platform (Sect. 4.5).

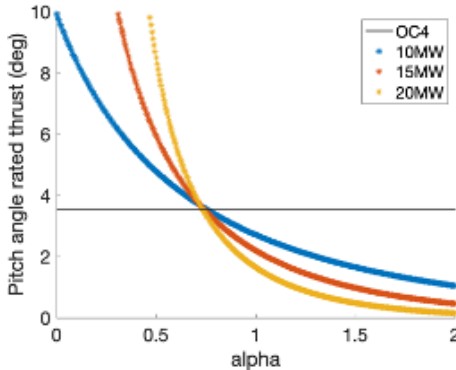

**Figure 9.** The OC4 platform pitch angle of upscaled systems at rated thrust. TS1

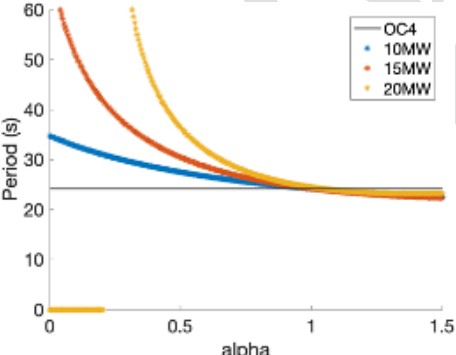

**Figure 10.** The OC4 natural period of upscaled systems.

The $T_{\text{n}}$ of the system is also evaluated using Eq. (8), shown in Fig. 10. The pitch $T_{\text{n}}$ is over 20 s for the entire range of $\alpha$ and is 24.2 s for the baseline OC4 5 MW system. Note that for the 20 MW upscaled system in the $\alpha = 0$–0.32 range, the system is unstable. This is because the platform stiffness term becomes negative as the center of mass of the system is raised, with a 20 MW turbine on a platform that is too small.

The semi-submersible platform is upscaled from the OC4 design using a scaling factor of $\alpha = 0.75$, which is the approximate value that preserves the static platform pitch angle at rated thrust. The results are shown in Table 9. The specific power, draft, clearance between the rotor and the waterline, wall thickness, and platform pitch angle are kept constant. The moment of inertia is shown for the entire system including the tower and RNA. The ratio of the platform steel mass to the total platform mass decreases from 27 % for the OC4 turbine to 18 % for the 20 MW upscaled system. Fitting a curve to the mass data indicates that the platform steel mass is upscaled by $R^{1.3}$, and the total platform mass is upscaled by $R^{1.8}$. The ballast mass increases more quickly than the steel mass, and the ballast mass is significantly cheaper. The natural period of the system in pitch increases slightly as it is upscaled.

**Table 9.** Upscaled OC4 table of results.

| Rated power | MW | 5 | 10 | 15 | 20 |
|---|---|---|---|---|---|
| Sp | $\mathrm{W\,m^{-2}}$ | 401 | 401 | 401 | 401 |
| $R$ | m | 63 | 89 | 109 | 126 |
| Draft | m | 20 | 20 | 20 | 20 |
| $\mathrm{CM_{platform}}$ | m | −13.6 | −13.1 | −12.5 | −12.0 |
| $\mathrm{CM_{system}}$ | m | −10 | −7.9 | −6.1 | −4.6 |
| Pitch angle | ° | 3.5 | 3.5 | 3.4 | 3.3 |
| Total stiffness | $\mathrm{Nm\,rad^{-1}}$ | 1.0E+09 | 2.7E+09 | 4.9E+09 | 7.6E+09 |
| $I_{\mathrm{system}}$ | $\mathrm{kg\,m^2}$ | 1.1E+10 | 3.5E+10 | 7.0E+10 | 1.2E+11 |
| Pitch natural period | s | 24.2 | 26.4 | 27.7 | 28.7 |
| Natural frequency | Hz | 0.26 | 0.24 | 0.23 | 0.22 |
| Steel mass | kg | 3.59E+06 | 5.60E+06 | 7.30E+06 | 8.86E+06 |
| Ballast mass | kg | 9.70E+06 | 2.0E+07 | 3.0E+07 | 4.0E+07 |
| Total platform mass | kg | 1.3E+07 | 2.5E+07 | 3.7E+07 | 4.9E+07 |
| Percent steel mass | | 27 % | 22 % | 20 % | 18 % |

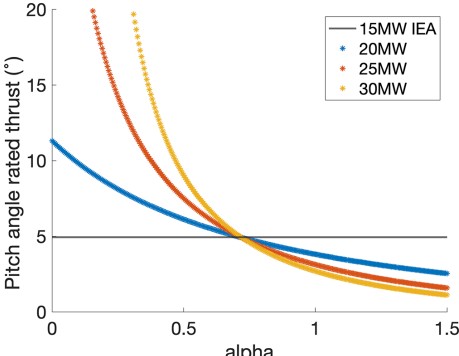

**Figure 11.** The IEA 15 MW platform pitch of upscaled systems at rated thrust.

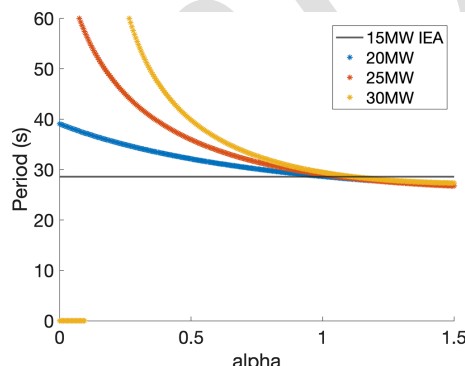

**Figure 12.** The IEA natural period of upscaled systems.

## 4.2 IEA platform upscaling results

Figure 11 shows the static platform pitch angle at rated thrust for each $\alpha$ increment for the IEA 15 MW upscaled platform. The platform dimensions increase with $\alpha$, and so the static platform pitch angle decreases. The platform pitch angle of the upscaled platforms matches the 15 MW IEA pitch value of 4.9° for the IEA 15 MW at $\alpha = 0.72$, similar to the OC4 upscaling. Further investigation of the upscaling factor is shown in the analytical model for the semi-submersible platform (Sect. 4.5).

The natural period is also calculated at each upscaling factor increment using Eq. (8). The pitch $T_{\mathrm{n}}$ for the 20, 25, and 30 MW is shown in Fig. 12. The pitch $T_{\mathrm{n}}$ is over 20 s for the entire upscaling range of $\alpha$, and it is 28.6 s for the baseline IEA 15 MW platform. Again, the platform becomes unstable for the 30 MW turbine in the $\alpha = 0$–0.18 range because of the high center of mass of the system with the relatively small platform.

The semi-submersible platforms are upscaled from the IEA design using a scaling factor of $\alpha = 0.72$ for the plat-

form dimensions, shown in Table 10. The 332 $\mathrm{W\,m^{-2}}$ specific power, 4.5 cm wall thickness, and 5.9° platform pitch angle are kept constant. Recall that the wall thickness is 4.5 cm based on the IEA design, in contrast to the 6 cm wall thickness of the OC4 design. The platform mass results would be significantly different if a larger wall thickness was used, if required for greater structural integrity. The ratio of the platform steel mass to the total platform mass is reduced as the turbines are upscaled; the IEA 15 MW system has 19 % steel mass, and the IEA 30 MW system has 19 % steel mass compared to the total platform mass including ballast. Fitting a curve to the mass data indicates that the platform steel mass is upscaled by $R^{1.4}$, and the total platform mass is also upscaled by $R^{2.2}$. $T_{\mathrm{n}}$ of the system in pitch increases slightly as it is upscaled.

## 4.3 Case study discussion

Upscaling of both platforms can be compared, specifically at 20 MW. Table 11 shows the comparison between the designs, including the lower specific power and larger rotor radius of the IEA 20 MW system. The upscaled IEA 20 MW platform

**Table 10.** Upscaled IEA 15 MW table of results.

| Rated power | MW | 15 | 20 | 25 | 30 |
|---|---|---|---|---|---|
| Sp | $W\,m^{-2}$ | 332 | 332 | 332 | 332 |
| Rotor radius | m | 120 | 138 | 155 | 170 |
| Draft | m | 20.0 | 22.1 | 24.0 | 25.7 |
| $CM_{platform}$ | m | −16.8 | −18.6 | −20.2 | −21.6 |
| $CM_{system}$ | m | −5 | −5.5 | −5.8 | −6.1 |
| Platform pitch | ° | 4.9 | 4.9 | 4.9 | 4.9 |
| Total stiffness | $Nm\,rad^{-1}$ | 3.3E+09 | 4.9E+09 | 6.8E+09 | 8.8E+09 |
| $I_{system}$ | $kg\,m^2$ | 5.3E+10 | 8.9E+10 | 1.4E+11 | 1.9E+11 |
| Pitch natural period | s | 28.6 | 30.3 | 31.9 | 33.2 |
| Natural frequency | Hz | 0.22 | 0.21 | 0.20 | 0.19 |
| Steel mass | kg | 3.5E+06 | 4.7E+06 | 6.0E+06 | 7.3E+06 |
| Seawater ballast mass | kg | 5.7E+06 | 7.7E+06 | 1.0E+07 | 1.2E+07 |
| Fixed ballast mass | kg | 9.2E+06 | 1.3E+07 | 1.6E+07 | 2.0E+07 |
| Total platform mass | kg | 1.8E+07 | 2.5E+07 | 3.2E+07 | 3.9E+07 |
| Percent steel mass | | 19 % | 19 % | 19 % | 19 % |

has a larger draft, smaller wall thickness, higher $CM_{system}$, and larger static platform pitch angle, whereas the upscaled OC4 20 MW platform has a larger stiffness, moment of inertia, steel mass, and total platform mass. The pitch natural period and ratio of platform steel mass to total steel mass are similar for both designs. The OC4 5 MW reference platform was designed in 2014 (Robertson et al., 2014), while the IEA 15 MW platform was designed in 2020 (Allen et al., 2020), which likely explains the reduction in the platform steel mass and wall thickness in the more recent design. The percentage of the platform steel mass relative to the total platform mass is relatively constant at 19 % for the IEA upscaling results (Table 10). In contrast, the percentage of the platform steel mass relative to the total platform mass decreases for the OC4 upscaling results (Table 9). Additionally, the IEA platform steel mass scales by $R^{1.4}$, while the OC4 platform steel mass scales by $R^{1.3}$. The IEA platform steel mass increases more rapidly in part because the draft is increasing, while the OC4 draft is constant.

There is a 30 m gap between the blade tip and waterline for both case studies, which was kept constant during upscaling. We chose the 30 m gap because of the prevalence of this choice in practice, but this clearance will need to be explored further in future research studies. In particular, the heave motion of each upscaled turbine should be considered to ensure that there is not too large of a downward motion towards the waterline while also considering wave height and combined platform rotational motions.

The case studies can be used to understand upscaling trends for floating platforms. Comparing four of the 5 MW OC4 systems with one upscaled 20 MW OC4 system, the total platform mass including ballast is similar, within 8 %; however the platform steel mass is reduced by up to 38 % for the single 20 MW turbine case. There is a lower ratio of the platform steel mass to the total platform mass for the

upscaled platforms, primarily because the wall thickness remains constant. Rotor power scales with $R^2$, and so turbine power will increase more rapidly than platform steel mass as the OC4 turbines are upscaled to the $R^{1.3}$. Additionally, the ballast mass does increase for the upscaled systems, but the ballast cost is likely significantly lower.

Comparing two 15 MW IEA systems with one upscaled 30 MW IEA system, the total platform mass including ballast is similar within 6 % of the mass of the 30 MW upscaled design. Additionally, the platform steel mass is 21 % lower for one 30 MW system as compared with two 15 MW systems. The IEA platform mass scales to $R^{1.4}$, and so the platform power will scale more quickly than steel mass. This result suggests that there are advantages to continued upscaling of turbines on floating platforms, specifically when the platform draft and wall thickness are kept constant.

It is notable that the static pitch angle of the platform varies for each case study design. The OC4 semi-submersible static pitch angle is 3.6°, and the IEA semi-submersible static pitch angle is 4.9°. Early FOWTs had a small static pitch angle to be conservative in design, but there are no absolute standards on what value of static pitch angle is acceptable.

The upscaling methodology is useful to identify trends for each platform type, but it should be noted that the designs are not being optimized. The original designs (OC4 and IEA 15 MW) are not optimized initially but are designed based on expertise. The upscaled designs are also not optimized, so it is possible that other platform designs may be more stable with less platform steel mass. Optimization studies can be conducted for individual projects at specific sites or for future research projects, but optimization is outside the scope of this research study. Future work will also estimate levelized cost of energy (LCOE) for the upscaled turbines.

**Table 11.** Comparison of the upscaled 20 MW IEA system with the upscaled 20 MW OC4 system.

|  |  | Upscaled IEA 20 MW | Upscaled OC4 20 MW |
|---|---|---|---|
| Sp | $W\,m^{-2}$ | 332 | 401 |
| Rotor radius | m | 138 | 126 |
| Draft | m | 22.1 | 20 |
| Wall thickness | m | 0.045 | 0.06 |
| $Dist_{cc}$ | m | 100 | 84 |
| $CM_{platform}$ | m | −16.7 | −11.7 |
| $CM_{system}$ | m | −5.0 | −4.8 |
| Pitch angle | ° | 4.9 | 3.5 |
| Total stiffness | $Nm\,rad^{-1}$ | 4.9E+09 | 7.6E+09 |
| $I_{system}$ | $kg\,m^2$ | 8.9E+10 | 1.2E+11 |
| Pitch natural period | s | 30.3 | 28.7 |
| Steel mass | kg | 4.8E+06 | 8.9E+06 |
| Seawater ballast mass | kg | 7.8E+06 | 4.0E+07 |
| Fixed ballast mass | kg | 1.2E+07 | 0 |
| Total platform mass | kg | 2.5E+07 | 4.9E+07 |
| Steel mass ratio |  | 19 % | 18 % |

## 4.4 Comparison of platform upscaling with similar studies for the OC4 platform

The upscaled OC4 semi-submersible design can also be compared to other semi-submersible upscaling studies (George, 2014; Leimeister et al., 2016; Kikuchi and Ishihara, 2019a). These upscaling studies do not seek to find platform scaling relations but instead upscale one specific design. As stated previously, Leimeister et al. (2016) upscale the OC4 to a 7.5 and 10 MW semi-submersible, George (2014) upscales the OC4 to a 10 MW semi-submersible, and Kikuchi and Ishihara (2019a, b) upscale the Fukushima FORWARD design to both a 7.5 and a 10 MW semi-submersible. Table 12 shows the 7.5 MW semi-submersible upscaling results, and Table 13 shows the 10 MW semi-submersible upscaling results. Both tables include the upscaled OC4 platform from this study.

All studies upscale the platform based on the increase in power rating. George (2014) and Kikuchi and Ishihara (2019a, b) limit certain dimensions such as draft and platform wall thickness, and all studies check criteria to ensure the design meets the natural period and static pitch angle requirements. Each of these studies use the RNA mass upscaling ratio in order to set the upscaling factor for the platform. Leimeister (2016) upscales the platform dimensions using a scaling factor of 1.264 for the 10 MW design, and then scaling is adjusted separately for the main column and upper columns. George (2014) uses a scaling factor of 1.26 for the 10 MW design, based on the mass scaling. For the 10 MW upscaling results, the other three studies all have a similar or smaller spread between the outer columns as compared with this study.

The platform dimensions of this study are within 3 % of the results of George (2014) for all platform dimensions shown. The only notable difference is that the calculated static pitch angle is lower for George (2014) even though the platform is slightly smaller than the one modeled in this study. The Leimeister et al. (2016) study is the only one that increases both the draft and the wall thickness with upscaling. Additionally, the platform pitch natural period is 34 % larger for Leimeister et al. (2016) due to the larger platform dimensions. Finally, the Kikuchi and Ishihara (2019a, b) study has a 16 % smaller distance between the outer columns, which causes an increase in the static pitch angle.

Overall, the Leimeister et al. (2016) study is the most conservative, the George (2014) study is the most similar to the method proposed here, and the Kikuchi and Ishihara (2019a, b) study increases the draft but reduces the spread between columns. The spread between the columns provides the largest contribution to stability for the semi-submersible platform type, so this reduction in column spread may have drawbacks.

This proposed upscaling method differs from the other methods in that there is one platform dimension upscaling factor identified, which can be used in Eq. (1) to upscale any semi-submersible platform. Note that the scaling relations are only valid for a similar semi-submersible design with three outer columns forming a triangle and one central turbine. This is in contrast to the other studies which upscale one specific case study through a variety of methods that include some trial and error and do not result in a scaling factor.

## 4.5 Analytical model for semi-submersible platform upscaling

There are classical analytical scaling laws for wind turbines (Manwell et al., 2009), but the scaling laws for FOWT platforms are not fully understood. The results from the case

**Table 12.** The 7.5 MW upscaled semi-submersible comparison.

|  |  | Leimeister (2016) | % difference from this study | George (2014) | % difference from this study | This study |
|---|---|---|---|---|---|---|
| Draft | m | 24.5 | 23 % | 20 | 0 % | 20 |
| Wall thickness | m | 0.078 | 29 % | 0.060 | 0 % | 0.060 |
| Rad$_{col}$ | m | 7.4 | 5 % | 6.8 | −3 % | 7.0 |
| Rad$_{hp}$ | m | 14.7 | 5 % | 13.6 | −3 % | 13.9 |
| Dist$_{cc}$ | m | 61.3 | 5 % | 56.5 | −3 % | 58.1 |
| Static pitch angle | ° | 3.7 | −6 % | 2.4 | −39 % | 3.9 |
| Pitch natural period | s | 34.1 | 34 % | 25.0 | −2 % | 25.5 |

**Table 13.** The 10 MW upscaled semi-submersible comparison.

|  |  | Leimeister (2016) | % difference from this study | George (2014) | % difference from this study | Kikuchi (2019) | % difference from this study | This study |
|---|---|---|---|---|---|---|---|---|
| Draft | m | 25.28 | 26 % | 20.0 | 0 % | 21.3 | 7 % | 20 |
| Wall thickness | m | 0.076 | 27 % | 0.060 | 0 % | 0.060 | 0 % | 0.060 |
| Rad$_{col}$ | m | 7.15 | −8 % | 7.6 | −3 % | 8.0 | 3 % | 7.8 |
| Rad$_{hp}$ | m | 15.17 | −2 % | 15.1 | −3 % | 16.0 | 3 % | 15.5 |
| Dist$_{cc}$ | m | 63.21 | −2 % | 63.0 | −3 % | 54.3 | −16 % | 64.8 |
| Static pitch angle | ° | 4.8 | 26 % | 3.1 | −18 % | 4.5 | 20 % | 3.8 |
| Pitch natural period | s | 33.2 | 26 % | 28.0 | 6 % | 26.0 | −1 % | 26.4 |

studies can be used to develop analytical upscaling relations for the semi-submersible platforms. For the static pitch angle to match the original semi-submersible design, an upscaling factor of approximately $\alpha = 0.72$–$0.75$ was found for both the OC4 5 MW and the IEA 15 MW designs. Additionally, the platform steel mass scales with $R^{1.3}$–$R^{1.4}$ when the wall thickness is kept constant. The platform mass would scale to a greater ratio if the wall thickness increases proportionally with $R$ (Sieros et al., 2012). Thus, the upscaling is more advantageous in terms of platform steel mass when the wall thickness is kept constant. However, increasing the platform wall thickness may be required for the structural integrity of the FOWT system, and this is an important area of future research. These results have been determined using hydrodynamic models and an iterative method, but fundamental equations for the static pitch when upscaling can also be derived analytically. The static pitch equation is shown in Eq. (11), which is an expanded version of Eq. (4).

$$\theta_p = \frac{M_{\text{aero}}}{\rho g W_{55} + \rho g V_{\text{disp}} \left( B - \text{CM}_{\text{system}} \right)} \tag{11}$$

The numerator is the aerodynamic moment on the platform at rated wind speed, which is the thrust of the wind turbine multiplied by the distance between the rotor hub and the center of mass of the system. The denominator of the equation is the platform stiffness $C_{55}^{\text{hydrostatics}}$. The mooring stiffness is neglected in these calculations; Sect. 4.3 addresses this assumption with a sensitivity study. The platform stiffness includes two terms, one is based on buoyancy $\left( V_{\text{disp}} \left( B - \text{CM}_{\text{system}} \right) \right)$ and the other on waterplane area ($W_{55}$). The waterplane area

term provides the dominant stability for semi-submersible platforms, contributing 94 % for the 15 MW IEA. The literature shows that the buoyancy term is always small for semi-submersible platforms, and other research studies have neglected the buoyancy term in the stiffness equation for semi-submersible upscaling (Kikuchi and Ishihara, 2019a). For both the OC4 and the IEA semi-submersible platforms, the buoyancy term is actually destabilizing because the turbine and tower mass raise the CM$_{\text{system}}$ above position $B$. Eq. (11) can be simplified to Eq. (12) by neglecting the buoyancy term and only considering the waterplane area term in the denominator.

$$\theta_{p,s} = \left[ \frac{\text{Th} \times h}{\rho g W_{55}} \right]_{\text{original}} = \left[ \frac{Th \times h}{\rho g W_{55}} \right]_{\text{new}} \tag{12}$$

There is aerodynamic similarity between the original and upscaled turbine with constant density of air, thrust coefficient, and rated wind speed. The thrust is Th, and $h$ is the distance between the hub and the CM$_{\text{system}}$. For simplicity, the hub height is used, and the distance from the CM$_{\text{system}}$ and the waterline is neglected. The hub height is 90 % of the total distance for the OC4 and 98 % for the IEA 15 MW designs. Additionally, the hub height is defined as 1.25 times the rotor radius in this model. This gives the 30 m clearance for the 15 MW IEA turbine and increases the clearance for larger turbines. The second moment area of the waterplane for the IEA semi-submersible is calculated using Eq. (13), which is

also shown in a simplified version in Eq. (14).

$$W_{55} = \frac{\pi}{4}\left[\text{Rad}_{\text{cent}}^4 + 3(\text{Rad}_{\text{col}})^4\right]$$
$$+ 2\pi(\text{Rad}_{\text{col}})^2(8.28 \cdot \sin 60 \cdot \text{Rad}_{\text{col}})^2, \tag{13}$$

$$W_{55} = \frac{\pi}{4}(\text{Rad}_{\text{cent}})^4 + \left(\frac{3\pi}{4}\right)(138)(\text{Rad}_{\text{col}})^4 \tag{14}$$

Calculating $W_{55}$ for the 15 MW IEA platform, the first term (central column) is 0.1 % of the total, and the second term (three outer columns) is 99.9 %, indicating that nearly all of the stability comes from the three outer columns. Thus, the first term is neglected, and only the column radius term is considered. The thrust and hub height equations are shown in Eq. (15). The rotor diameter is defined as $\phi$, with $\phi_{\text{original}}$ as the original rotor diameter and $\phi_{\text{new}}$ as the rotor diameter of the upscaled turbine.

$$\frac{\left(\frac{1}{2}\rho_a \pi \left(\frac{\phi_{\text{original}}}{2}\right)^2 C_T u^2\right)\left(1.25 \cdot \left(\frac{\phi_{\text{original}}}{2}\right)\right)}{103.6\pi \left(\text{Rad}_{\text{col\_original}}\right)^4}$$
$$= \frac{\left(\frac{1}{2}\rho_a \pi \left(\frac{\phi_{\text{NEW}}}{2}\right)^2 C_T u^2\right)\left(1.25 \cdot \left(\frac{\phi_{\text{NEW}}}{2}\right)\right)}{103.6\pi \left(\text{Rad}_{\text{col\_NEW}}\right)^4} \tag{15}$$

Note that any semi-submersible with three outer columns forming an equilateral triangle would reduce to the same equation because the coefficient terms cancel out. Eq. (15) can be further simplified to only include the column radius and rotor diameter, as shown in Eq. (16).

$$\frac{\left(\phi_{\text{original}}\right)^3}{\left(\text{Rad}_{\text{col\_original}}\right)^4} = \frac{(\phi_{\text{NEW}})^3}{\left(\text{Rad}_{\text{col\_NEW}}\right)^4} \tag{16}$$

This scaling relation for the semi-submersible platform can determine the column radius needed for an upscaled semi-submersible platform based on the original column radius and the diameter of the original and upscaled turbines. Equation (17) is in a similar format to the generic scaling relation shown in Eq. (1). The scaling factor between the upscaled column radius and the original column radius is $\alpha = 0.75$, which is very similar to the upscaling factor of $\alpha = 0.72$–0.75 that was found for the semi-submersible case studies. Thus, the analytical formulation recovers the same upscaling factor as the more complex hydrodynamic model.

$$\text{Rad}_{\text{col\_NEW}} = \left(\text{Rad}_{\text{col\_original}}\right) \cdot \left[\frac{\phi_{\text{NEW}}}{\phi_{\text{original}}}\right]^{3/4} \tag{17}$$

This relation is similar to the square–cube law of blade upscaling, except that it shows that platform upscaling is likely to be advantageous because platform stiffness scales faster than the wind turbine overturning moment. The upscaled column radius scales at $\alpha = 0.75$ because the overturning moment from rated thrust is proportional to the diameter cubed,

and the stiffness is dominated by the column radius to the fourth power. This only defines column radius and column spread, but all parameters can be upscaled by the same $\alpha$ of 0.75 for a semi-submersible upscaled design. Additionally, if it is assumed that all semi-submersible platform dimensions increase, including wall thickness and draft, the platform steel mass increases by a factor of 2.25 in Eq. (18). However, if the platform wall thickness is kept constant, as it was in the case studies, the platform steel mass increases by a factor of 1.5. If multiple small FOWTs were used instead of upscaling, the steel mass would scale as $R^2$ because the power would scale with the rotor swept area and the system mass would scale proportionally (Manwell et al., 2009; Jamieson, 2018).

$$M_{\text{platform\_NEW}} = \left(M_{\text{platform\_original}}\right) \cdot \left[\frac{\phi_{\text{NEW}}}{\phi_{\text{original}}}\right]^{1.5} \tag{18}$$

## 4.6   Sensitivity studies

The results presented above rely on a variety of assumptions, which are now assessed using parameter sensitivity studies.

### 4.6.1   Mooring line sensitivity

This research assumes that the stiffness contributions from the mooring lines can be neglected for the first-order platform pitch angle calculations. A mooring line sensitivity study is conducted to evaluate the contribution of mooring line stiffness to platform pitch motion. The study uses the OC4 system using OpenFAST and evaluates the natural period of the system when the mooring line stiffness is reduced. The published $T_n$ of the system is 27 s. OpenFAST is run with the tower degrees of freedom off, and the pitch $T_n$ is calculated using a free decay test (Figs. 13–14). The mooring line stiffness (EA) is then decreased from the original stiffness value to a stiffness that is one-eighth of the original value. $T_n$ is calculated for each simulation to determine the impact on the system dynamics. Table 14 shows that reducing the mooring line stiffness by a factor of 8 reduces the pitch $T_n$ of the system by less than 1 %.

The static pitch angle at rated thrust can also be evaluated in OpenFAST while decreasing the mooring line stiffness. OpenFAST is run for the OC4 system at steady, rated wind speed (Fig. 15). Table 15 shows that when the mooring line stiffness is reduced by a factor of 8, the static pitch angle is increased by less than 1 %. Thus, while the mooring design may change as the platform size increases, these results indicate that the mooring stiffness has a negligible impact on the platform dynamics and so can be ignored in upscaling studies. Further analysis of mooring line behavior is therefore outside the scope of this research.

**Table 14.** Pitch natural period of OC4 with reduced mooring line stiffness.

| EA (MN) | 753.6 | 502.4 | 376.8 | 188.4 | 94.2 |
|---|---|---|---|---|---|
| Tn (s) | 25.535 | 25.5625 | 25.590 | 25.645 | 25.740 |

**Table 15.** Static pitch angle of OC4 with mooring stiffness.

| EA (MN) | 753.6 | 502.4 | 376.8 | 188.4 | 94.2 |
|---|---|---|---|---|---|
| Pitch angle (°) | 3.2592 | 3.2618 | 3.2636 | 3.2729 | 3.2809 |

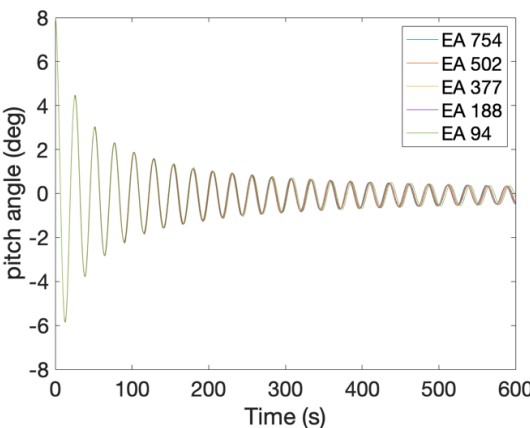

**Figure 13.** Free decay test for OC4.

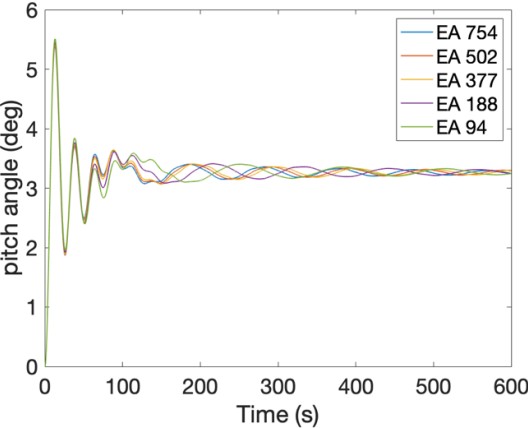

**Figure 15.** Static pitch of OC4 at rated thrust with mooring line stiffness.

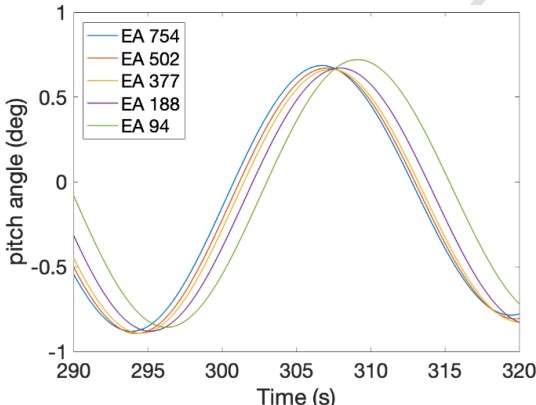

**Figure 14.** Free decay test of OC4 for a 30 s interval.

### 4.6.2 Rotor nacelle assembly mass sensitivity

The upscaling case studies are conducted with an assumption about the mass scaling of the rotor nacelle assembly. Classical upscaling uses $R^3$ scaling of RNA mass, while scaling based on data results in $R^{2.2}$ scaling approximately. In addition to the impact on turbine mass, the change in RNA mass scaling also impacts the FOWT platform design, mass, and cost. While some researchers have focused on reducing turbine mass to reduce the size and cost of the platform (Ward et al., 2021), it is still unclear if RNA mass reduction is a major design driver for FOWT design.

An RNA mass sensitivity study is conducted for the IEA 15 MW wind turbine. The RNA mass is reduced to 50 % of the original mass, while the tower mass remains constant. As the RNA mass decreases, the platform size is reduced so that the platform pitch angle at rated thrust remains constant. Table 16 shows the results of the RNA mass sensitivity study for the 15 MW IEA semi-submersible.

The results show that the $CM_{system}$ is lowered with the reduced RNA mass. A smaller platform is also needed to support the smaller RNA mass (because of the lower center of mass). The platform steel mass is reduced by 8 % (300 000 kg) when the RNA mass is reduced by 50 % (510 000 kg), and the total platform mass including ballast is reduced by 10 %. The waterplane area component of the platform stiffness is stabilizing, while the contribution from the center of mass is destabilizing; thus, when the RNA mass is reduced by 50 %, the destabilizing stiffness term is reduced by 35 %. The RNA mass impacts the upscaling results, but the sensitivity study shows that it is reasonable to assume the constant $R^{2.2}$ RNA upscaling within the scope of this study.

**Table 16.** RNA mass sensitivity results for IEA 15 MW.

| RNA mass reduction | RNA mass | Platform pitch angle | CM | Total stiffness | Steel mass | Steel mass reduction | Total platform mass | Platform mass reduction |
|---|---|---|---|---|---|---|---|---|
| | kg | ° | m | Nm rad$^{-1}$ | kg | % | kg | % |
| Original | 1.02E+06 | 7.8 | −2.7 | 2.77E+09 | 3.88E+06 | | 1.84E+07 | |
| 5 % | 9.66E+05 | 7.6 | −3.1 | 2.86E+09 | 3.88E+06 | 0 % | 1.85E+07 | 0 % |
| 15 % | 8.64E+05 | 7.8 | −3.4 | 2.81E+09 | 3.81E+06 | −2 % | 1.79E+07 | −3 % |
| 29 % | 7.19E+05 | 7.7 | −4.1 | 2.85E+09 | 3.73E+06 | −4 % | 1.75E+07 | −5 % |
| 42 % | 5.87E+05 | 7.7 | −4.8 | 2.87E+09 | 3.65E+06 | −6 % | 1.70E+07 | −7 % |
| 50 % | 5.09E+05 | 7.9 | −5.1 | 2.81E+09 | 3.58E+06 | −8 % | 1.65E+07 | −10 % |

## 4.7   Discussion of results

The results are useful for upscaling a semi-submersible platform to a larger size, especially as a preliminary design analysis before a more detailed design process. These results are applicable to a semi-submersible platform with three outer columns forming a triangle and the turbine mounted in the center. There are a wide variety of other FOWT designs that would be interesting to study, including more unique semi-submersible designs (e.g., four outer columns forming a square with one central column or the turbine mounted on one outer column instead of the central column), spar designs, and tension leg platform designs. If a researcher wants to upscale a triangular semi-submersible platform with three outer columns and the turbine centrally mounted to a size of 6–30 MW, this method can give a good estimate of the platform dimensions and mass based on an original design and larger wind turbine parameters.

The limitations of this method include several of the simplifications made to identify upscaling trends. The dynamics of the FOWT system need further evaluation, including second-order effects. However, this study chooses to focus exclusively on the platform pitch motion during rated thrust, as this has been shown to be an important load case. Additionally, environmental conditions such as wind–wave misalignment are not considered in this case. The purpose of this research study is to identify the upscaling trends using simplified assumptions and leave further evaluation of detailed design to future research studies. The benefit of this method is identifying an upscaled design with little computational time and expense.

Future work should validate the upscaled FOWT designs using OpenFAST, which involves creating a turbine and platform model for each upscaled design. Additionally, future research is needed to assess the structural integrity of the FOWT platform assuming constant wall thickness with upscaling. The constant clearance assumption between the blade tip and waterline can also be assessed, in addition to checking the heave motions to ensure that they are within a reasonable range. A better understanding of the upscaled designs in extreme wind and wave conditions can further the knowledge of platform upscaling. An additional area of future work is to conduct cost of energy analysis to gain insight into how turbine and platform scaling impact the system economics. Upscaling the platform with a constant wall thickness causes the platform steel mass to increase with a factor of approximately $R^{1.5}$, suggesting that larger turbines may be advantageous. But a more nuanced and detailed analysis is needed, which includes balancing system costs and estimates on annual energy production, to assess the likely impact of continued upscaling of FOWTs.

## 5   Conclusion

Floating offshore wind turbines are being developed to harness energy in windy, deep-water sites. While individual floating platform designs can be optimized for a specific site, this research provides fundamental insight that can guide technological development by creating a generalized methodology for semi-submersible platform upscaling. This work has resulted in an upscaling factor for a triangular semi-submersible platform with three outer columns and a centrally mounted turbine. The upscaling factors for dimensions and mass are comparable to the classical turbine scaling relations (Manwell et al., 2009).

The numerical method used in the methodology was validated using OpenFAST. The upscaled platform results are the closest to those of George (2014), but the results do differ from other similar research studies (Kikuchi and Ishihara, 2019a; Leimeister et al., 2016; George, 2014; Ju et al., 2020). Additionally, this study differs from the generic semi-submersible scaling study conducted by Wu and Kim (2021) because their method is an iterative approach to find the column radius and spread for a semi-submersible, and there is no scaling factor provided.

Two upscaling case studies are evaluated: the OC4 semi-submersible turbine is upscaled from 5 to 20 MW, and the IEA 15 MW semi-submersible turbine is upscaled from 15 to 30 MW. The semi-submersible scale factor for both case studies is approximately $\alpha = 0.75$, using both numerical and analytical methods. These relations can be used to quickly estimate the platform dimensions for a larger turbine rotor. Additionally, the analytical solution shows that the platform

steel mass increases with $R^{1.5}$ when the platform wall thickness is kept constant using the upscaling method and $R^2$ when multiple, smaller FOWTs are used instead of upscaling. Thus, platform upscaling is shown to be advantageous regarding platform steel mass cost savings as compared to installing multiple, smaller FOWT systems. Having fewer, larger FOWT systems will improve other aspects of offshore wind farms, such as fewer turbines to install and maintain in difficult-to-access ocean environments. However, there will likely need to be an upper limit to FOWT upscaling, likely related to the increased stresses due to blade weight that continue to scale linearly with the rotor radius. The upscaling of FOWT systems is already taking place in industry, and a better understanding of platform scaling can give key insight into research and industry development.

**Code and data availability.** The code used can be accessed at https://github.com/Kayliemct/upscaling_fowt (Roach and Lackner, 2023). This includes the MATLAB upscaling code for both the IEA and OC4 platforms. The data plots are also included, as well as excel files from OpenFAST free decay tests.

**Author contributions.** CE1 KLR and MAL developed the upscaling methodology together. KLR developed the code and did the upscaling analysis. MAL supervised KLR throughout this time and provided valuable advice and input in the coding and method. KLR wrote the draft version of this paper, and both MAL and JFM reviewed and edited the draft version. JFM gave valuable feedback during the revisions and throughout the research project.

**Competing interests.** The contact author has declared that none of the authors has any competing interests.

**Disclaimer.** Publisher's note: Copernicus Publications remains neutral with regard to jurisdictional claims made in the text, published maps, institutional affiliations, or any other geographical representation in this paper. While Copernicus Publications makes every effort to include appropriate place names, the final responsibility lies with the authors.

**Acknowledgements.** We thank Mareike Leimeister and Nataliia Sergiienko for taking the time to thoroughly review this paper and discuss with us.

**Review statement.** This paper was edited by Amy Robertson and reviewed by Mareike Leimeister and Nataliia Sergiienko.

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

## Remarks from the language copy-editor

CE1    Please verify the section.

## Remarks from the typesetter

TS1    We have to forward the requested replacement of the figures to the handling editor of the paper for approval together with a brief explanation. Please send me a brief statement about why these two figures are different and need to be replaced. I will then take care of the following processes. Many thanks.

TS2    Please provide date of last access.