# Peer review of "A New Methodology for Upscaling Semi-submersible Platforms for Floating Offshore Wind Turbines"

_Wind Energy Science, 2023_

## Referee Comment (RC1)

**General comments:**

The paper addresses the scaling trends of semi-submersible type floating offshore wind turbine support structures, considering the trend of further increasing wind turbine MW-classes. The presented approach is quite simplistic; however, relevant aspects are touched on in some more detailed discussion and sensitivity studies. Overall, the work presents a valuable insight into the development of larger floating support structures, which is a suitable basis for further research and future detailed investigations on floating wind turbine structures and economics.
Overall, the paper is well written (apart from some minor mistakes mentioned under technical corrections). The simplistic approach might leave some room for more investigations and detailed discussion (noted under specific comments).

**Specific comments:**

- "generalized upscaling relations that can be used for other semi-submersible platforms" (Abstract) or "This study is the first to develop generalized upscaling relations for semi-submersible FOWT platforms" (1 Introduction) or "to upscale any semi-submersible platform" (4.4)
    - These formulations are a bit misleading, based on which something more generalized is expected by the reader.
    - Semi-submersibles are much more different in terms of their designs as just the two very similar floaters considered in this case study. A significant difference, for example, would have been a design with just three columns and the turbine on one edge of the triangle or a design with four columns.
    - These are too ambitious formulations. The study is still considering only a specific semi-submersible floater design. Due to the high similarity of the investigated two designs, it is questionable if the found scaling factors are applicable to other semi-submersible designs in general.
    - Please add a discussion on the "universality" of your approach, as there are so many different design solutions for semi-submersibles, which are not covered by the two systems considered in this case study.
- Assuming constant values – Need of further elaborations and discussion. Where are limits of this approach, keeping the draft and especially the wall thickness constant?
    - Keeping the wall thickness constant. – Is this a realistic and feasible approach, considering that the diameters of the columns might increase? Has this approach been checked wrt structural integrity? It is furthermore striking that the wall thickness for the IEA-based designs (thus, the larger ones) are even kept constant at a smaller value than those of the OC4-based design. Is this suitable? The resulting numbers (lines 319-321) might change significantly if the wall thickness is changed as well if this is required for structural integrity reasons. This last aspect, however, is addressed later in your discussion. Maybe it can be pointed to this already here in lines 319-321, when presenting the results.
    - Using a constant value for the gap between bottom of rotor plane and water line. – Is this a reasonable approach? In your upscaling approach, you only keep the pitch motion constant, but not the heave motion as well.
- Chapter 1 (Introduction), lines 33-34: I would add here, what size is already addressed by the industry, e.g. 18 MW (https://www.rechargenews.com/wind/ge-has-18mw-offshore-wind-turbine-giant-in-the-works-vernova-chief-strazik/2-1-

1418184?utm_source=email_campaign&utm_medium=email&utm_campaign=2023-03-13&utm_term=recharge&utm_content=daily).

- Chapter 2 (Background):
  - o The literature review might be extended. It is not at all addressed the huge diversity of semi-submersible designs and related different upscaling approaches.
  - o The scaling approach by Leimeister is not only applied to obtain a 7.5 MW design but also to obtain a 10 MW design. This was part of another publication (M. Leimeister, E.E. Bachynski, M. Muskulus, and P. Thomas, 2016. 'Design Optimization and Upscaling of a Semi-Submersible Floating Platform'. *Proceedings of the WindEurope Summit 2016, September 27-29, 2016, Hamburg, Germany*.). But both information is also contained in the Master Thesis available at: http://resolver.tudelft.nl/uuid:7f6b5eda-15d8-4228-ad9a-8c27f8c5c258.
- Chapter 3 (Methodology):
  - o Figure 3: X-/surge is mainly directed by the main wind direction. This is not always equal to the wave direction. The information in the figure is a bit misleading.
  - o You are talking about validation (Section 3.3 – line 249). This is, however, just a verification. This is just a verification. And the large discrepancies would not directly lead to the conclusion that the model is verified (and not at all validated). However, the approximate results might be sufficient for the focus and application in this study. – Please rephrase and at least add some justification.
  - o The approach needs to be presented in some more detail. Furthermore, please elaborate on why this approach is followed. Such a root-finding problem is not really needed for finding the value of alpha that results in equal rated platform pitch angles, as you have the underlying equations, based on which you can determine the required scaling constant, what you also did later in the paper. This is, however, furthermore not a new approach, as this was already done in (Leimeister, 2016). What are the differences and maybe advantages of this approach compared to the direct calculation? Right now I see a disadvantage of having more simulations/calculations with this root-finding problem approach. Please elaborate on this, maybe in Section 4.5.
  - o You mention that the pitch natural period is calculated and checked that it is not in the predominant wave period range. How large do you define this range? And what is done if the pitch natural period would be close to the predominant wave period range?
- Chapter 4 (Results and Discussion)
  - o The structure is not clear at the very beginning. When reading, the reader thinks about some shortcomings, which, however, are later on discussed.
  - o Please refer at certain points (e.g., 4.1 and 4.2 when presenting the results for alpha) to further discussions done (e.g., 4.5 comparison to the analytical calculations).
  - o Tables 9 and 10: Please discuss on the different trends in the percent steel mass for OC4 and IEA systems.
  - o Section 4.4 and Table 13: Please include the results from Leimeister, 2016 for the 10 MW upscaled design and correct the information in line 395: No constant scaling factor of 1 is used for the entire platform. There are different scaling factors used for different parts of the floater (main column and upper columns).
  - o Good investigations and discussion in Section 4.5.
  - o Great to have a sensitivity study. However, the final resume of the second sensitivity study (4.6.2) is missing.
- Chapter 5 (Conclusion): The shortcomings and outlook might be extended and elaborated on in more detail. This might then be moved to a separate section before the conclusion. Please also add a discussion on the "universality" of your approach, as there are so many different design

solutions for semi-submersibles, which are not covered by the two systems considered in this case study.

**Technical corrections:**

- Throughout the paper, please write parameters (both within the text and in equations) in math environment/formula style.
- Please simplify your equations. There are very often brackets used where no brackets are needed.
- Please write out parameter descriptions (referring to the third and fourth line in Table 5).
- For reasons of consistency, please write 8° instead of 8 deg in line 232.
- Line 232: "of 25.5 s" should rather be "is 25.5 s".
- Line 243: "36% error is" should rather be "36% error in".
- Line 268: "which is increased in from 0 to 2 in increments of 0.005". There seems to be something wrong. I would delete the first "in".
- Line 291: The draft has a unit of "m" and not "MW".
- Line 376: Missing full stop after "research study".
- Line 513: Delete the "In" at the very beginning of this sentence.
- Lines 519/520: Use intext citation.

---

## Author Comment (AC1)

Dear Referee #1 and Referee #2,

Thank you both for your comments and feedback. This document provides responses to each comment and makes note of changes made in our paper.

**Reply to comments received on 16 March 2023 from Referee #1**

1. The paper addresses the scaling trends of semi-submersible type floating offshore wind turbine support structures, considering the trend of further increasing wind turbine MW-classes. The presented approach is quite simplistic; however, relevant aspects are touched on in some more detailed discussion and sensitivity studies. Overall, the work presents a valuable insight into the development of larger floating support structures, which is a suitable basis for further research and future detailed investigations on floating wind turbine structures and economics.
Overall, the paper is well written (apart from some minor mistakes mentioned under technical corrections). The simplistic approach might leave some room for more investigations and detailed discussion (noted under specific comments).

**Reply**: Thank you for your feedback. Yes, our goal is to provide a general approach that can be applied to multiple semi-submersible platforms of different dimensions. This is meant to aid in understanding upscaling trends, and can be combined with more extensive analysis for a specific offshore wind turbine system.

**Specific Comments**
2. "generalized upscaling relations that can be used for other semi-submersible platforms" (Abstract) or "This study is the first to develop generalized upscaling relations for semi-submersible FOWT platforms" (1 Introduction) or "to upscale any semi-submersible platform" (4.4)
o These formulations are a bit misleading, based on which something more generalized is expected by the reader.
o Semi-submersibles are much more different in terms of their designs as just the two very similar floaters considered in this case study. A significant difference, for example, would have been a design with just three columns and the turbine on one edge of the triangle or a design with four columns.
o These are too ambitious formulations. The study is still considering only a specific semi-submersible floater design. Due to the high similarity of the investigated two designs, it is questionable if the found scaling factors are applicable to other semi-submersible designs in general.
o Please add a discussion on the "universality" of your approach, as there are so many different design solutions for semi-submersibles, which are not covered by the two systems considered in this case study.

**Reply**: Yes, there is still a wide variety of floating offshore wind turbine designs, and types of semi-submersibles. However, this paper is focusing on semi-submersible platforms with three outer columns, and the turbine in the center of the triangle. The justification for this focus is

because both FOWT reference semi-submersibles have this general design (5 MW OC4 semi-submersible and 15 MW IEA semi-submersible). There are some design differences (such as the pontoons in the IEA semi-submersible), but the scaling trends are still the same for both. We are not claiming that the same scaling relations would hold for a unique semi-submersible design, such as with four columns, or three columns with the turbine mounted in one corner.

**Revision to paper:**

- changes to the abstract are highlighted in yellow:

**Abstract.** This paper presents a new upscaling methodology for semi-submersible floating offshore wind turbine platforms. The size and power rating of offshore wind turbines have been growing in recent years, with modern wind turbines rated at 10 - 14 MW in contrast with 2 - 5 MW in 2010. It is not apparent how much further wind turbines can be increased before it is unjustified. Scaling relations are a useful method for analyzing wind turbine designs, to understand the mass, load, and cost increases with size. Scaling relations currently do not exist but are needed for floating offshore platforms to understand how the technical and economic development of floating offshore wind energy may develop with increasing turbine size. In this paper, a hydrodynamic model has been developed to capture the key platform response in pitch. The hydrodynamic model is validated using OpenFAST, a high-fidelity offshore wind turbine simulation software. An upscaling methodology is then applied to two semi-submersible case studies of reference systems (5 MW OC4 and 15 MW IEA). For each case study, the platform pitch angle at rated wind turbine thrust is constrained to a specified value. The results show that platform dimensions scale to a factor of 0.75, and the platform steel mass scales to a factor of 1.5 when the wall thickness is kept constant. This study is the first to develop generalized upscaling relations that can be used for other semi-submersible platforms that have three outer columns with the turbine mounted at the center of the system. This is in contrast with other studies that upscale a specific design to a larger power rating. This upscaling methodology provides new insight into trends for semi-submersible platform upscaling as turbine size increases.

- changes to the introduction are highlighted in yellow:

model is validated using OpenFAST, a high-fidelity offshore wind turbine simulation software (Jonkman, 2019; National Renewable Energy Laboratory, 2020). The methodology is then applied using two semi-submersible case studies, in which the platform pitch angle at rated wind turbine thrust is constrained to a constant value. Other researchers have upscaled specific semi-submersible platforms (George, 2014; Leimeister *et al.*, 2016; Ju *et al.*, 2020; Kikuchi and Ishihara, 2020). This study is the first to develop generalized upscaling relations for semi-submersible FOWT platforms with three outer columns and a central turbine. Ideally, this research would also be conducted with other types of semi-submersible designs, as well as other

FOWT designs (spar, tension leg platform). These upscaling relations can provide new insight into design trends for three column semi-submersible platforms as turbine size increases. Additionally, the paper identifies key underlying physics behind the semi-submersible upscaling relations.

- changes to 4.4 "Comparison of Platform Upscaling with Similar Studies for the OC4 Platform" are highlighted in yellow.

410     the draft but reduces the spread between columns. The spread between the columns provides the largest contribution to stability for the semi-submersible platform type, so this reduction in column spread may have drawbacks.

This proposed upscaling method differs from the other methods in that there is one platform dimension upscaling factor identified, which can be used in Eq. (1) to upscale any semi-submersible platform. Please note that the scaling relations are only valid for a similar semi-submersible design with three outer columns and one central turbine. This is in contrast to the

- changes to the conclusion:

**5 Conclusion**

Floating offshore wind turbines are being developed to harness energy in windy, deep-water sites. While individual floating platform designs can be optimized for a specific site, this research provides fundamental insight that can guide technology

520     development by creating a generalized methodology for semi-submersible platform upscaling. This work has resulted in an upscaling factor for three column semi-submersible platforms with a central turbine. The upscaling factors for dimensions and mass that is comparable to the classical turbine scaling relations (Manwell, McGowan and Rogers, 2009).

The numerical method used in the methodology was validated using OpenFAST. The upscaled platform results are closest to those of George (George, 2014), but the results do differ from other similar research studies (George, 2014; Leimeister *et al.*,

525     2016; Kikuchi and Ishihara, 2019a; Ju *et al.*, 2020). Additionally, this study differs from the generic semi-submersible scaling study conducted by Wu and Kim (Wu and Kim, 2021), because their method is an iterative approach to find the column radius and spread for a semi-submersible, and there is no scaling factor provided.

3. Assuming constant values – Need of further elaborations and discussion. Where are limits of this approach, keeping the draft and especially the wall thickness constant?
o Keeping the wall thickness constant. – Is this a realistic and feasible approach, considering that the diameters of the columns might increase? Has this approach been checked wrt structural integrity? It is furthermore striking that the wall thickness for the IEA-based designs (thus, the larger ones) are even kept constant at a smaller value than those of the OC4-based design. Is this suitable? The resulting numbers (lines 319-321) might change significantly if the wall thickness is changed as well if this is required for structural integrity reasons. This last aspect, however, is addressed later in your discussion. Maybe it can be pointed to this already here in lines 319-321, when presenting the results.
o Using a constant value for the gap between bottom of rotor plane and water line. – Is this a reasonable approach? In your upscaling approach, you only keep the pitch motion constant, but not the heave motion as well.

**Reply**: Keeping semi-submersible platform draft constant is reasonable, because based on the literature, the semi-submersible designs tend to grow in column spread faster than they do draft. For example, both the 5 MW OC4 semi-submersible and the 15 MW IEA semi-submersible had the same draft of 20 meters.
- Significant structural analysis of the platform designs was outside the scope of this research. The authors were also surprised that the 15 MW IEA platform had a 4.5 cm wall thickness, while the 5 MW OC4 design had the larger 6 cm wall thickness. We chose to

keep a constant wall thickness while upscaling, but the results would vary if the wall thickness were increased during upscaling.

- The authors chose to keep the 30 meter clearance between the blade tip and the waterline, as opposed to scaling the hub height in another way. The literature and industry trends suggest that the 30 meter clearance is typical for offshore wind turbines, and so we wanted to use what we believe is a realistic design based on current trends. The platform pitch angle at rated thrust was kept constant because literature has shown that this parameter often governs the extreme loads of FOWT systems. We chose to limit the platform pitch angle for our simplified model. Your suggestion to include the heave motion as a part of our model would be valuable for the next iteration.

**Revision to paper:**
- changes to 4.1 "OC4 Platform Upscaling Results"

wall thickness, and platform pitch angle are kept constant. Please recall that the wall thickness is 4.5 cm based on the IEA design, in contrast to the 6 cm wall thickness of the OC4 design. The platform mass results would be significantly different if a larger wall thickness was used. The moment of inertia is shown for the entire system including the tower and RNA. The ratio

325     of platform steel mass to total platform mass decreases from 27% for the OC4 turbine to 18% for the 20 MW upscaled system. Fitting a curve to the mass data indicates that the platform steel mass is upscaled by $R^{1.3}$ and the total platform mass is upscaled by $R^{1.8}$. The ballast mass is increasing more quickly than the steel mass, and ballast mass is significantly cheaper. The natural period of the system in pitch increases slightly as it is upscaled.

4. Chapter 1 (Introduction), lines 33-34: I would add here, what size is already addressed by the industry, e.g. 18 MW (https://www.rechargenews.com/wind/ge-has-18mw-offshore-wind-turbine-giant-in-the-works-vernova-chief-strazik/2-1-1418184?utm_source=email_campaign&utm_medium=email&utm_campaign=2023-03-13&utm_term=recharge&utm_content=daily).

**Revision to paper:**
- changes to Introduction

Offshore wind turbine size and capacity have been growing rapidly over the past ten years as well. Modern offshore wind turbines designed by General Electric, Siemens Gamesa, and Vestas have ratings of 10 - 18 MW with blade diameters exceeding 200 m (Siemens Gamesa, 2020; GE Renewable Energy, 2023; Vestas, 2023). GE has only recently announced their

35     18 MW turbine design (Buljan, 2023). In contrast, in 2010 offshore turbines had power ratings between 2 - 5 MW and blade diameters were 75 - 125 m (Musial *et al.*, 2022). Even larger designs are likely to be developed in the future, with researchers even investigating a 50 MW turbine (Yao *et al.*, 2021).

5. Chapter 2 (Background):
o The literature review might be extended. It is not at all addressed the huge diversity of semi-submersible designs and related different upscaling approaches.
o The scaling approach by Leimeister is not only applied to obtain a 7.5 MW design but also to obtain a 10 MW design. This was part of another publication (M. Leimeister, E.E. Bachynski, M. Muskulus, and P. Thomas, 2016. 'Design Optimization and Upscaling of a Semi-Submersible Floating Platform'. *Proceedings of the WindEurope Summit 2016, September 27-*

*29, 2016, Hamburg, Germany*.). But both information is also contained in the Master Thesis available at: http://resolver.tudelft.nl/uuid:7f6b5eda-15d8-4228-ad9a-8c27f8c5c258.

**Reply**: The wide variety of semi-submersible designs is outside the scope of this paper. The 10 MW upscaling Leimeister et al. results have been added, thank you for bringing this to our attention.

**Revision to paper:**
- changes to 4.4 "Comparison of Platform Upscaling with Similar Studies for the OC4 Platform"

385 **4.4 Comparison of Platform Upscaling with Similar Studies for the OC4 Platform**

The upscaled OC4 semi-submersible design can also be compared to other semi-submersible upscaling studies (George, 2014; Leimeister *et al.*, 2016; Kikuchi and Ishihara, 2019a). These upscaling studies do not seek to find platform scaling relations, but instead upscale one specific design. As stated previously, Leimeister et al. (Leimeister *et al.*, 2016) upscales the OC4 to a 7.5 MW and 10 MW semi-submersible, George (George, 2014) upscales OC4 to a 10 MW semi-submersible, and Kikuchi and

390 Ishihara (Kikuchi and Ishihara, 2019a, 2019b) upscale the Fukushima FORWARD design to both a 7.5 MW and 10 MW semi-submersible. Table 12 shows the 7.5 MW semi-submersible upscaling results and Table 13 shows the 10 MW semi-submersible upscaling results. Both tables include the upscaled OC4 platform from this study.

**Table 13: 10 MW Upscaled Semi-submersible Comparison**

|  |  | Leimeister (2016) | % difference from this study | George (2014) | % difference from this study | Kikuchi (2019) | % difference from this study | This study |
|---|---|---|---|---|---|---|---|---|
| Draft | m | 25.28 | 26% | 20.0 | 0% | 21.3 | 7% | 20 |
| Wall thickness | m | 0.076 | 27% | 0.060 | 0% | 0.060 | 0% | 0.060 |
| $Rad_{col}$ | m | 7.15 | -8% | 7.6 | -3% | 8.0 | 3% | 7.8 |
| $Rad_{hp}$ | m | 15.17 | -2% | 15.1 | -3% | 16.0 | 3% | 15.5 |
| $Dist_{cc}$ | m | 63.21 | -2% | 63.0 | -3% | 54.3 | -16% | 64.8 |
| Static pitch angle | deg | 4.8 | 26% | 3.1 | -18% | 4.5 | 20% | 3.8 |
| Pitch natural period | s | 33.2 | 26% | 28.0 | 6% | 26.0 | -1% | 26.4 |

All studies upscale the platform based on the increase in power rating. George (George, 2014) and Kikuchi and Ishihara

400 (Kikuchi and Ishihara, 2019a, 2019b) limit certain dimensions such as draft and platform wall thickness, and all check criteria

6. Chapter 3 (Methodology):
o Figure 3: X-/surge is mainly directed by the main wind direction. This is not always equal to the wave direction. The information in the figure is a bit misleading.
o You are talking about validation (Section 3.3 – line 249). This is, however, just a verification. This is just a verification. And the large discrepancies would not directly lead to the conclusion that the model is verified (and not at all validated). However, the approximate results

might be sufficient for the focus and application in this study. – Please rephrase and at least add some justification.

o The approach needs to be presented in some more detail. Furthermore, please elaborate on why this approach is followed. Such a root-finding problem is not really needed for finding the value of alpha that results in equal rated platform pitch angles, as you have the underlying equations, based on which you can determine the required scaling constant, what you also did later in the paper. This is, however, furthermore not a new approach, as this was already done in (Leimeister, 2016). What are the differences and maybe advantages of this approach compared to the direct calculation? Right now I see a disadvantage of having more simulations/calculations with this root-finding problem approach. Please elaborate on this, maybe in Section 4.5.

o You mention that the pitch natural period is calculated and checked that it is not in the predominant wave period range. How large do you define this range? And what is done if the pitch natural period would be close to the predominant wave period range?

**Reply**: Yes, the authors agree that there can be wind, wave misalignment. The figure is not trying to say that wind and waves are always present in the surge direction, but is primarily showing the system degrees of freedom. Wind and waves can be offset by an angle from the surge direction.

- Sect. 3.3 has been changed from "validation" to "verification".
- Regarding the root finding approach, while it may be possible to solve for a single alpha value analytically to satisfy the constraint, the root finding approach is preferred for this study. This approach produces trends for the platform behavior (figures 9 - 12), which provides insight into how the upscaling constant alpha impacts the platform response. We can see the tradeoffs between a more conservative and less conservative design more clearly.
- The DNV-GL report "Environmental Conditions and Environmental Loads" (DNVGL-RP-C205) suggests that the pitch natural period of a semi-submersible platform should always be above 20 seconds. If the pitch natural period were below 20 seconds for any cases, then we would re-design.

**Revision to paper:**
- changes to 3.3 "Verification of the Hydrodynamic Model for Case Study Turbines"

250   for the 5 MW OC4 system, but not published for the IEA 15 MW system. The platform pitch angle from the hydrodynamic model can be used as a relative rather than absolute pitch angle in order to constrain the upscaled turbine platform pitch angle. The pitch natural period from the OpenFAST free decay test (Figure 8) is estimated as 27.7 s, the published value is 29.5 s, and the result from the hydrodynamic model is 28.6 s. The model has a 3% error relative to the OpenFAST results and a 3% error relative to the published value. The verification results are summarized in Table 8. The purpose of this verification was

255   to confirm that the calculations were similar to both published values as well as OpenFAST simulations. The model could be further verified with other simulation software or with data from FOWT pilot projects, but further validation is outside the scope of this paper.

7. Chapter 4 (Results and Discussion)

o The structure is not clear at the very beginning. When reading, the reader thinks about some shortcomings, which, however, are later on discussed.

o Please refer at certain points (e.g., 4.1 and 4.2 when presenting the results for alpha) to

further discussions done (e.g., 4.5 comparison to the analytical calculations).
o Tables 9 and 10: Please discuss on the different trends in the percent steel mass for OC4 and IEA systems.
o Section 4.4 and Table 13: Please include the results from Leimeister, 2016 for the 10 MW upscaled design and correct the information in line 395: No constant scaling factor of 1 is used for the entire platform. There are different scaling factors used for different parts of the floater (main column and upper columns).
o Good investigations and discussion in Section 4.5.
o Great to have a sensitivity study. However, the final resume of the second sensitivity study (4.6.2) is missing.

**Reply**: Please see the changes made below for your comments.

- For section 4.6.2 "Rotor Nacelle Assembly Mass Sensitivity Study" the sensitivity study was only conducted for the IEA 15 MW turbine and not for the 5 MW OC4 turbine. This is because the 15 MW turbine has a greater mass which would be more impacted by the scaling factor chosen as it increases to 30 MW.

**Revision to paper:**
- changes to 4 "Results and Discussion"

   **4 Results and Discussion**

[revised manuscript text omitted]

8. Chapter 5 (Conclusion): The shortcomings and outlook might be extended and elaborated on in more detail. This might then be moved to a separate section before the conclusion. Please also add a discussion on the "universality" of your approach, as there are so many different design solutions for semi-submersibles, which are not covered by the two systems considered in this case study.

**Reply**: Thank you for your suggestion, please see new section 4.7 "Discussion of Results"

**Revision to paper:**
- addition of section 4.7 "Discussion of Results"

**4.7 Discussion of Results**

530 The results are useful for upscaling a semi-submersible platform to a larger size, especially as a preliminary design analysis before a more detailed design process. These results are applicable for a semi-submersible platform with three outer columns and the turbine mounted in the center. There are a wide variety of other FOWT designs that would be interesting to study, including more unique semi-submersible designs (e.g., four columns or the turbine mounted on one outer column), spar designs, and tension-leg platform designs. If a researcher wants to upscale a three column semi-submersible platform to a size
535 of 6 - 30 MW, this method can give a good estimate of the platform dimensions and mass based on an original design and larger wind turbine parameters.

**Technical corrections:**
9. Throughout the paper, please write parameters (both within the text and in equations) in math environment/formula style.

**Reply**: Changes made throughout the paper, including the following.

**Revision to paper:**
- Section 3.1 "Hydrodynamic Modeling of Floating Platforms"

The stability of a FOWT can be characterized using the hydrodynamic loading and response. Eq. (2), known as the Cummins equation, is the equation of motion for an offshore platform in water with six degrees of freedom (TU Delft, 2006; Jonkman, 2007; Duarte, Sarmento and Jonkman, 2014). $M_{ii}$ is the mass or mass moment of inertia term, $A_{ii}$ is the added mass coefficient

140 term, $K_{ii}$ is the retardation matrix, and $C_{ii}$ is the stiffness matrix. The platform acceleration, velocity, and displacements are represented by $\ddot{q}^{tot}$, $\dot{q}^{tot}$, and $q^{tot}$ respectively, $F_i^{waves}$ is the external wave loading, $F_i^{rotor}$ is the force of the wind turbine acting on the floating platform, and $h_i$ is the moment arm of $F_i^{rotor}$ for rotational platform degrees of freedom. The six degrees of freedom are labeled with $i = 1,2,...6$ and correspond to (surge, sway, heave, roll, pitch, yaw). Figure 3 shows the FOWT coordinate system (Sebastian and Lackner, 2012).

10. Please simplify your equations. There are very often brackets used where no brackets are needed.

**Revision to paper:**

[revised manuscript text omitted]

2. Introduction, line 35 - after 2009, it was a number of studies proving that the wind turbine mass does not scale as 'square cube' but closer to square or even less.

**Reply:**
- Yes, we discuss the theory of the "square cube" law as well as the reality of different scaling related to technological innovation over time, starting on line 87.

Historical data from wind turbines of different sizes can also be used to understand upscaling trends. For example, historical data indicates that the rotor mass has increased to the power of between 2 and 2.5, not the cubic power of the square-cube law (Figure 1) (Jamieson, 2018). This smaller value for the scaling exponent is primarily due to technological innovation, such as

90    new materials and improved manufacturing, in newer designs that are usually larger in size (Jamieson, 2018; Shields *et al.*, 2021).

3. Introduction, line 40 - LCOE of offshore wind is not twice of onshore, please refer to the most recent data

**Reply:**
- I have updated the LCOE numbers based on the literature, rather than approximating double the cost.

**Revision to Paper:**

the number of installed units in a wind farm for a given total capacity, which is motivated by the large per unit cost (including

40    the foundation, installation, electrical interconnection, and maintenance visits at sea). Offshore wind levelized cost of energy (LCOE) is approximately $84/MWh for fixed offshore wind, $58 - $120/MWh for floating offshore wind, and onshore wind is $37.8/MWh (Musial *et al.*, 2022; U.S. Energy Information Administration, 2022). As offshore wind energy development continues, it is important to understand if even larger turbines can continue to reduce the LCOE of offshore wind farms, or if there is an upper limit to the cost effectiveness and practicality of upscaling.

4. The authors cannot claim that their paper is "the first", please refer to https://doi.org/10.1016/j.rser.2022.112477 and similar papers.

**Reply:**

- This paper is very helpful, thank you for sharing. However, this paper reviews semi-submersible platforms 5 - 15 MW in order to understand upscaling trends. In contrast, this paper provides an upscaling study based on two case study reference turbines to suggest a general upscaling relationship for semi-submersible platforms.

**Revision to Paper:**

distance between the columns.

110    Sergiienko et al. (Sergiienko *et al.*, 2022) has reviewed semi-submersible platforms of 5 - 15 MW in order to understand upscaling trends. The results indicate that the wind turbine mass scales with the square of the rotor diameter. Additionally, the authors find that most platform developers keep the draft constant at 20 m for FOWTs in the 5 - 15 MW range (Sergiienko *et al.*, 2022).

When upscaling a FOWT, specific load cases are typically used to constrain the design and ensure acceptable stability and

5. In-text referencing should be improved - for example, Leimeister et al. (Leimeister et al., 2016) - should be Leimeister et al. (2016)

**Reply:**

- The authors are following the in-text referencing guidelines as outlined by the journal template.

**Abstract.** Please use only the styles of this template (MS title, Authors, Affiliations, Correspondence, Normal for your text, and Headings 1–3). Figure 1 uses the style Caption and Fig. 1 is placed at the end of the manuscript. The same is applied to tables (Smith et al., 2014; Miller and Carter, 2015) adipiscing elit. Mauris dictum, nibh ut condimentum pharetra, quam ligula varius est, sed vehicula massa erat ut metus. In eget metus lorem. Fusce vitae ante dictum, elementum sem non, lacinia

10    dui.

6. Equation (2) - the authors have not included viscous damping acting on the platform

**Reply:**

- Yes, the authors are neglecting viscous damping for this research study as it does not impact the static platform pitch at rated wind speed.

7. Equations (3) and the following ones should be referenced

**Reply:**

- Yes, these equations are referenced from Delhommeau, 1993, "19th WEGEMT School Numerical Simulation of Hydrodynamics: Ships and Offshore Structures"

8. The authors should make clear where is the origin of their coordinate system, is it the centre of gravity of waterline?

**Reply:**
- The origin of the coordinate system is the waterline.

**Revision to Paper:**

140 represented by $\ddot{q}^{tot}$, $\dot{q}^{tot}$, and $q^{tot}$ respectively, $F_i^{waves}$ is the external wave loading, $F_i^{rotor}$ is the force of the wind turbine acting on the floating platform, and $h_i$ is the moment arm of $F_i^{rotor}$ for rotational platform degrees of freedom. The six degrees of freedom are labeled with $i = 1,2,...6$ and correspond to (surge, sway, heave, roll, pitch, yaw). Figure 3 shows the FOWT coordinate system with the center of the coordinate system at the waterline (Sebastian and Lackner, 2012).

$$(M_{ii} + A_{ii})\ddot{q}^{tot} + \int_0^T K_{ii}(t - \tau)\dot{q}^{tot}(\tau)d\tau + C_{ii}q^{tot} = F_i^{waves} + F_i^{rotor} * h_i \qquad (2)$$

9. The authors use a lot of figures taken from other papers/reports. Please make sure that you have obtained all copyrights to use them in your work.

**Reply:**
- We will work with the editorial team at Wind Energy Science to determine the appropriate attribution of the figures.

10. Section 3.2.1 - not clear why the authors decided to upscale the wind turbines if the solutions already exist for 10 and 15 MW?

**Reply:**
- The authors chose to upscale the 5 MW OC4 platform to 10, 15, and 20 MW because we wanted to compare the upscaling results with existing 10 and 15 MW semi-submersible platforms. In sect. 4.3 "Case Study Discussion" we discuss the different upscaling results from the OC4 platform as compared with the IEA platform.

11. Section 3.3 - it is not clear how exactly the authors have modelled Equation (4). Also, how the authors obtained hydrodynamic parameters, and state-space model of the radiation force for their model?

**Reply:**
- The authors are solving for the platform pitch angle by using the thrust force, distance from the system center of gravity to the hub height, and the platform stiffness.

12. The reviewer is a bit concerned by the 10% error in natural periods, because to check natural period, you just need to make sure that the mass, added mass and hydrostatic stiffness are correct, and all these numbers are available in the public domain.

**Reply:**

- The natural period of the OC4 system is calculated using OpenFAST and also using the hydrodynamic model (which is based on the calculated moment of inertia and platform stiffness). Both the results from OpenFAST and the calculated results are lower than the natural period based on the platform pitch natural frequency published in the OC4 platform report.

13. In Table2, the authors mentioned the CM as -13.46 - but this number is for the platform itself without installed wind turbine. When assembling OC4 with a WT, you will get -9.9 m or so, have the authors taken this into account in their model? Also, the mass matrix will be changed.

**Reply:**

- Yes, the authors have included the center of mass for the tower and RNA. The total system CM is -9.9 m for the 5 MW OC4 platform.

14. Figures 5, and 6 - the authors need to demonstrate the comparison between OpenFAST and their model

**Reply:**

- Figures 5 and 6 are results from OpenFAST runs conducted by the authors. The numerical root finding approach does not include non-linear, time domain models. The root finding approach provides an estimate for platform pitch angle and natural period which are discussed.

15. Line 245 - error is 36% is not acceptable, the authors should have contacted the authors of UMaine to get more accurate data. Also, it is possible to calculate the assembled mass matrix if the platform and wind turbine are known

**Reply:**

- TBD The authors did contact the authors from UMaine to get clarification on the 15 MW IEA data. There is no published value for the platform pitch angle for the 15 MW IEA system. The 36% error is between the 3.6º platform pitch angle that the authors obtained from running OpenFAST and the 4.9º platform pitch angle that the authors obtained using the numerical calculations. The authors from UMaine stated that the platform pitch angle of the system based on hydrostatic stiffness only is 7.1º.

16. Section 3.4 - it is not clear why the authors decided to apply the scaling factor to both radius and distance, as the product of these two is correlated to the hydrostatic stiffness. Please provide more explanation on this step

**Reply:**

- The authors did not consider scaling the product of radius and distance, instead of scaling both parameters. In equation 7, the stiffness is proportional to (column radius)^4 and (column radius)^2*(distance between columns)^2. By scaling both column radius and distance with alpha, the stiffness is scaled as alpha^4 and both terms impact the stiffness equally. If we scaled the product of (column radius)*(distance between the columns), then the terms may scale differently from each other.

17. Line 275, have the authors assumed that the rated wind speed is the same for all wind turbines?

**Reply:**
- No, the rated wind speed is based on the case study. The rated wind speed for the OC4 5 MW is 11.4 m/s and the rated wind speed for the IEA 15 MW is 10.59 m/s.

18. The authors have not mentioned a possibility of peak shaving to decrease the wind turbine max thrust force if needed to the platform design purposes.

**Reply:**
- The maximum thrust force is our conservative approach based on the published baseline system properties. Control modifications could limit the maximum thrust force in practice.

19. "The added mass coefficient cA comes from the documentation for each semi-submersible case study" - have the authors scaled this parameter as well or used fixed for all platform dimensions?

**Reply:**
- The authors have kept the added mass coefficient as a fixed parameter for all upscaled platforms in the study. The added mass coefficient cA was 0.63 for both the 5 MW OC4 platform and the 15 MW IEA platform.

20. The authors numerically found that their scaling factor for linear dimensions is 0.75. This result can be obtained also using some manual calculations. Say you have R - distance to the column, and d - diamater of the column, hydrostatic stiffness scales as $R^2 d^2$. Also, Thurst force scales as $D^2$, hub height as D. Stiffness = Force x height/angle, so stiffness should scale as $D^3$. If we have stiffness = $D^3$ and at the same time $R^2 d^2$, so (Rd) should scale as $D^{(3/2)} = D^{1.5}$. If we scale both R and d at the same time, so each of them will scale as $D^{(1.5/2)} = D^{(0.75)}$.

**Reply:**
- The authors found that the platform parameters (column radius, spacing, etc) scale as 0.75 using both a root finding method (Sect. 4.1 - 4.2) analytical calculations (Sect. 4.5).

21. Please revisit all your results and support by simple analysis referring to fundamental equations of the platform stiffness, etc.

**Reply:**
- The authors have aimed to provide fundamental equations in Sect. 3 "Methodology." Additionally, we have provided both a numerical and an analytical analysis in Sect. 4 "Results." Please provide more detail on what you are suggesting.

22. As was found in one of the review papers, the anticipated linear scaling for combined radius and distance parameter is 1.5 (0.75+0.75), but most recent designs are close to 1 due to applied optimisation of each particular design.

**Reply:**
- The authors have found a scaling factor of 0.75 based on analytical and numerical methods. It is possible that existing FOWT systems have not followed this exact trend, especially as new designs are both larger and more technologically advanced and optimized. The results in this study are useful for determining upscaling trends for a FOWT platform prior to an optimization study.

---

## Referee Report (RR1)

**General comments:**

The authors have overall well addressed the reviewer's comments and provided valuable answers. There are just some minor comments and recommendations left for further improvement of the paper to high quality.

**Specific comments:**

- Please use a clear term for calling your semi-submersible platform type. For example, "three column semi-submersible platform" is misleading, since you consider semi-submersibles with four columns, of which three are forming the edges of a triangle and the central fourth is supporting the turbine.
  In this context, it would make sense to adjust the phrasing in Section 4.7 (line 535) as well, as you have already considered a semi-submersible with four columns but you may want to address a semi-submersible with four columns forming a square.
- Further comments on assuming constant values:
  - Keeping the wall thickness constant
    - The added sentences "Please recall that the wall thickness is 4.5 cm based on the IEA design, in contrast to the 6 cm wall thickness of the OC4 design. The platform mass results would be significantly different if a larger wall thickness was used." in lines 331-333 (section 4.1 on the OC4 Platform Upscaling Results – and not on the IEA design) rather fit to line 356 in 4.2 on the IEA Platform Upscaling Results. Please move/adjust.
    - The sentence "The platform mass results would be significantly different if a larger wall thickness was used." in lines 332/333 might be adjusted by adding, for example, "… if required for structural integrity reasons".
    - Please add (in both cases) a reference to section 4.5, where you address the influence of the wall thickness on the platform steel mass in more detail.
    - Please mention the relevance of structural integrity. Even this is out of the scope of your study, this should be highlighted that this needs to be considered carefully in the next research work and for more detailed design approaches.
  - Regarding the constant value for the gap between bottom of rotor plane and water line, I fully agree that taking a value of 30 m which is common for offshore wind turbines is a reasonable approach. My only concern is that this value accounts for sufficient clearance wrt extreme waves but not specifically for an additionally reduced gap due to any heave motion of a floating wind turbine system – i.e. in a bad circumstance, the wind turbine might just have a negative heave motion (and maybe some additional pitch) when a huge wave crest is hitting the structure. If you – as you are writing – agree that it would be valuable to include the heave motion in the next iteration, maybe you can just point it out as recommendation. There might not be the need to have the heave motion as a direct "design" parameter, however, it should be at least checked at the end, how the upscaled system performs in this degree of freedom.
- Chapter 3 (Methodology):
  - Please include your explanation and reasoning for using the root finding approach just as you explain it in your answers also in the paper (e.g. in 3.4). This is really valuable to understand the advantages and your intention for using this approach.

- o Please add the information (and reference to the recommended practice) you provide in your answers on the pitch natural period and predominant wave period as well in the paper, e.g. in line 284.
- Chapter 4 (Results and Discussion)
  - o For the reader it would be easier to understand where you get the numbers from, if you could include references to tables 10 and 9 in lines 370 and 371, respectively.
  - o Please correct the numbers for the OC4 platform steel mass scaling, as you have written $R^{1.3}$ in Section 4.1 (line 335), Section 4.3 (line 380), and Section 4.5 (line 438), but $R^{1.2}$ in Section 4.3 (line 372).
  - o For reasons of clarity, please mention that the added numbers (highlighted text) in lines 412 to 416 refer to the 10 MW upscaled designs (and not to the 7.5 MW ones).
  - o Please add in line 414 to complete the sentence: "…, and then scaling is adjusted separately for the main column and the upper columns."
  - o Please add some final resume to the second sensitivity study (4.6.2).
  - o Thanks for adding the discussion section 4.7. I would highly recommend to add there also the shortcomings you have found and partially already addressed beforehand, as well as some outlook on the next steps and future research. You might use some of the detailed elaboration in the last paragraph of the conclusion and then reduce the outlook then in the conclusion itself.

**Technical corrections:**

- References: Please make sure that the new reference (Leimeister, 2016) is used at the right places, i.e. in contrast to the reference (Leimeister et al., 2016) which does not cover the upscaling to the 10 MW wind turbine.
  - o Thus, in lines 399/400, the reference (Leimeister, 2016) would need to be added.
  - o The intext citation in line 4.13 needs to be corrected to just Leimeister (i.e. remove "et al.") to match the correctly used reference (Leimeister, 2016).
  - o And please add the 10 MW upscaling application (Leimeister, 2016) as well in section 2 (line 99).
- Please check again throughout the entire paper (both within the text and tables as well as equations) that all mathematical parameters are written in math environment/formula style and that this is also only done for mathematical parameters and not also for abbreviations (e.g. RNA). For example, there are parameters not yet in math environment in Equations 1 (R), 2 - 5, 7, Tables 2, 5, 9 - 16, and line 497 (EA).
- Some equations could still be further simplified by removing brackets that are not needed, e.g. Equations 2, 4, 5, 8, 11.
- Please ensure throughout the paper, that the same table style is applied, i.e. in some tables the header is bolt in others not, and other tables do not have a header at all.
- Abstract, line 8: For reasons of consistency, it might be worth updating the modern wind turbine rating number to 18 MW (instead of 14 MW) as well.
- Please introduce the abbreviation "NREL" when mentioning the full name in line 133.
- Please ensure that numbers and the corresponding unit are kept together and not separated over two lines (as, for example, in lines 174/175). However, this might change again when the final format is applied.
- Line 179: I think that, based on the journal's guidelines, "section 4.6.1" should rather start with a capital "S".

- The abbreviation RNA is already introduced in line 203. Thus, there is no need to introduce it again in lines 226 and 516. Furthermore, why is RNA in lines 203, 412 and from line 516 on in the text (Section 4.6.2 and Table 16) written in italics? It is not a parameter but just an abbreviation.
- For reasons of consistency, please write the number 1025 in line 290 as well with a comma at the thousands (1,205).
- Line 291: Please add the unit (m) after the number -13.46.
- Lines 316 and 436: Please correct "upscale factor" into "upscaling factor".
- Please check the journal guidelines, whether Section should always be written out. I just wondered as you have always written Section, but now used Sect. in lines 319 and 344.
- Table 9:
  - There is no need to introduce the parameter "R" again for the rotor radius.
  - Why are the entries in the first line (apart from the one in the first column) bolt?
- Line 435: There seems to be a "to" missing in "can be used develop".
- Please introduce the parameter Tn in the text before using it in Table 14.
- Lines 543/544: Please correct the grammar of this sentence. "that is" may just be replaced by "are".

---

## Author Response (AR2)

Thank you for your detailed review of our paper. We appreciate your suggestions for areas of further improvement and clarification. Please see our changes listed below.

**Reply to specific comments**

1. The term for the semi-submersible platform type has been changed throughout the paper, from "three column semi-submersible" to "triangular semi-submersible platform with three outer columns and the turbine centrally mounted" or simply the "triangular semi-submersible."

**4.7 Discussion of Results**

The results are useful for upscaling a semi-submersible platform to a larger size, especially as a preliminary design analysis before a more detailed design process. These results are applicable for a semi-submersible platform with three outer columns

535 forming a triangle and the turbine mounted in the center. There are a wide variety of other FOWT designs that would be interesting to study, including more unique semi-submersible designs (e.g., four outer columns forming a square with one central column or the turbine mounted on one outer column instead of the central column), spar designs, and tension-leg platform designs. If a researcher wants to upscale a triangular semi-submersible platform with three outer columns and the turbine centrally mounted to a size of 6 - 30 MW, this method can give a good estimate of the platform dimensions

540 and mass based on an original design and larger wind turbine parameters.

2.1 Changes have been made regarding the constant wall thickness assumption.

The semi-submersible platforms are upscaled from the IEA design using a scaling factor of $\alpha$ = 0.72 for the platform

355 dimensions, shown in Table 10. The 332 W/m$^2$ specific power, 4.5 cm wall thickness, and 5.9º platform pitch angle are kept constant. Please recall that the wall thickness is 4.5 cm based on the IEA design, in contrast to the 6 cm wall thickness of the OC4 design. The platform mass results would be significantly different if a larger wall thickness was used, if required for greater structural integrity. The ratio of platform steel mass to total platform mass is reduced as the turbines are upscaled; the 15 MW IEA system has 19% steel mass, and the 30 MW IEA system has 19% steel mass compared to total platform mass

360 including ballast. Fitting a curve to the mass data indicates that the platform steel mass is upscaled by $R^{1.4}$ and the total platform mass is also upscaled by $R^{2.2}$. The natural period of the system in pitch increases slightly as it is upscaled.

2.2 That is an excellent point regarding the constant 30 meter gap assumption. This is now briefly mentioned in section 3.4 "upscaling methodology" and 4.3 "case study discussion."

the scaling constant $\alpha$ in Eq. (1), which is increased from 0 to 2 in increments of 0.005. The wall thickness and clearance between the blade tip and the waterline are kept constant during upscaling. The 30 m clearance between the rotor and the waterline was chosen because the literature and industry trends show that the 30 m clearance is typical for offshore wind

280 turbines to date (Robertson, A., Jonkman, J., Masciola, M., Song, 2014; Allen et al., 2020). The system mass, buoyancy, ballast

value that preserves the static platform pitch angle at rated thrust. The results are shown in Table 9. The specific power, draft,

335 clearance between the rotor and the waterline, wall thickness, and platform pitch angle are kept constant.  The moment of inertia is shown for the entire system

380    increases more rapidly in part because the draft is increasing while the OC4 draft is constant.

There is a 30 m gap between the blade tip and waterline for both case studies, which was kept constant during upscaling. We chose the 30 m gap because of the prevalence of this choice in practice, but this clearance will need to be explored further in future research studies. In particular, the heave motion of each upscaled turbine should be considered to ensure that there is not too large of a downward heave motion towards the waterline in any case, to ensure that the gap is not too small considering

385    wave height and combined platform rotational motions.

**3.1 An explanation of the choice of the root-finding method has been added to the section 3.4 "upscaling methodology" section.**

This method is effectively a root-finding problem to determine the value of $\alpha$ that results in equal rated platform pitch angles for the baseline and upscaled turbines. While it may be possible to solve for a single alpha value analytically, the root-finding

275    approach was selected because it allows us to see trends for the platform behavior. We can clearly see how the upscaling value of $\alpha$ would result in a more conservative or less conservative design. The platform dimensions are upscaled uniformly with

**3.2 The reference has been added here.**

Eq. (4). The pitch natural period is calculated using Eq. (8) (derived from Eq. (5)) to ensure that it is not in the predominant wave period range. The pitch natural period of a semi-submersible platform should always be above 20 s (Det Norske Veritas

290    Germanischer Lloyd, 2017). The added mass coefficient $c_A$ comes from the documentation for each semi-submersible case

**4.1**

of platform steel mass relative to the total platform mass is relatively constant at 19% for the IEA upscaling results (Table 10). In contrast, the percentage of platform steel mass relative to the total platform mass decreases for the OC4 upscaling results

380    (Table 9). Additionally, the IEA platform steel mass scales by $R^{1.4}$ while the OC4 platform steel mass scales by $R^{1.32}$. The

**4.2 All mention of the OC4 steel mass upscaling now are consistently $R^{1.3}$.**

380    (Table 9). Additionally, the IEA platform steel mass scales by $R^{1.4}$ while the OC4 platform steel mass scales by $R^{1.32}$. The

**4.3 & 4.4**

425    to ensure the design meets natural period and static pitch angle requirements. Each of these studies use the *RNA* mass upscaling ratio in order to set the upscaling factor for the platform. Leimeister et al. (Leimeister, 2016) upscales the platform dimensions using a scaling factor of 1.264 for the 10 MW design, and then scaling is adjusted separately for the main column and upper columns. This is the starting point, and then scaling is adjusted for the main column. George (George, 2014) uses a scaling factor of 1.26 for the 10 MW design, based on the mass scaling. For the 10 MW upscaling results, the other three studies all

**4.5 There is a brief conclusion added to the end of the 4.6.2 sensitivity study section.**

when the *RNA* mass is reduced by 50%, the destabilizing stiffness term is reduced by 35%. The *RNA* mass impacts the upscaling results, but the sensitivity study shows that it is reasonable to assume the constant $R^{2.2}$ *RNA* upscaling within the

550    scope of this study.

**4.6 The suggested changes have been made to both section 4.7 "discussion of results" and section 5 "conclusion."**

560 The limitations of this method include the simplifications assumed in order to identify the upscaling trends. The dynamics of the FOWT system needs further evaluation, including second order effects. However, this study chooses to focus exclusively on the platform pitch motion during rated thrust, as this has been shown to be the primary load case. Additionally, environmental conditions such as wind wave misalignment are not considered in this case. The purpose of this research study is to identify the upscaling trends using the simplified assumptions, and leave further evaluation of detailed design to future

565 research studies. The benefit of this method is identifying an upscaled design with little computational time and expense. Future work should validate the upscaled FOWT designs using OpenFAST, which involves creating a turbine and platform model for each upscaled design. Additionally, future research is needed of the structural integrity of the FOWT platform assuming constant wall thickness with upscaling. The constant clearance assumption between the blade tip and waterline would also be beneficial, in addition to checking the heave motions of future research studies to ensure that heave motions are within

570 a reasonable range for platform motions. A better understanding of the upscaled designs in extreme wind and wave conditions

can further the knowledge of platform upscaling. An additional area of future work is to conduct cost of energy analysis, in order to gain insight into how turbine and platform scaling impact the system economics. Upscaling the platform with a constant wall thickness causes the platform steel mass to increase with a factor of approximately $R^{1.5}$, suggesting that larger turbines may be advantageous. But a more nuanced and detail analysis is needed, which includes balance of system costs and

575 estimates on annual energy production, to assess the likely impact of continued upscaling of FOWTs.

are used instead of upscaling. Thus, platform upscaling is shown to be advantageous regarding platform steel mass cost savings as compared to installing multiple, smaller FOWT systems. Having fewer, larger FOWT systems will improve other aspects

595 of offshore wind farms, such as fewer turbines to install and maintain in difficult to access ocean environments. However, there will likely need to be an upper limit to FOWT upscaling, likely related to the increased stresses due to blade weight that continue to scale linearly with rotor radius.

600

**Reply to technical corrections**

- The Leimeister reference has been corrected in the three places that you mention.
- All notation and abbreviations have been checked.
- The equations have been simplified by removing brackets.
- All tables have been checked so that size 9 cambria math font is used and there are no bold headers. The information is shown across rows rather than down columns, except in places where information is changing in rows and columns (tables 1 (from reference), 8-13, 16). The color has been removed from table 8.
- The modern wind turbine rating has been updated in the abstract.
- The abbreviation for NREL has been introduced.
- The numbers and units are together, and not split between lines.
- The mention of "Section 4.6.1" has been capitalized.
- The rotor nacelle assembly is now only defined once, on line 203.
- The comma has been added to 1,025 kg/m$^2$.

- The unit has been added to- 13.46 m.
- "Upscaling factor" has been corrected.
- The abbreviation "Sect." should be used within the text, I have modified places that read "Section". This is now in line with the journal's guidelines.
- There is no longer bold used in table 9, or any other table. Rotor radius is no longer introduced in table 9.
- The "to" has been added in line 449.
- The natural period abbreviation "Tn" is now introduced in line 231, and used throughout the paper.
- The grammatical mistake on line 581 has been changed.